JCB Journal of Cell Biology

# Kinetochore-bound Mps1 regulates kinetochore–microtubule attachments via Ndc80 phosphorylation

Krishna K. Sarangapani[1]*, Lori B. Koch[2,3,4]*, Christian R. Nelson[2,3]*, Charles L. Asbury[1], and Sue Biggins[2,3]

Dividing cells detect and correct erroneous kinetochore–microtubule attachments during mitosis, thereby avoiding chromosome missegregation. The Aurora B kinase phosphorylates microtubule-binding elements specifically at incorrectly attached kinetochores, promoting their release and providing another chance for proper attachments to form. However, growing evidence suggests that the Mps1 kinase is also required for error correction. Here we directly examine how Mps1 activity affects kinetochore–microtubule attachments using a reconstitution-based approach that allows us to separate its effects from Aurora B activity. When endogenous Mps1 that copurifies with kinetochores is activated in vitro, it weakens their attachments to microtubules via phosphorylation of Ndc80, a major microtubule-binding protein. This phosphorylation contributes to error correction because phospho-deficient Ndc80 mutants exhibit genetic interactions and segregation defects when combined with mutants in other error correction pathways. In addition, Mps1 phosphorylation of Ndc80 is stimulated on kinetochores lacking tension. These data suggest that Mps1 provides an additional mechanism for correcting erroneous kinetochore–microtubule attachments, complementing the well-known activity of Aurora B.

## Introduction

The equal segregation of duplicated chromosomes to daughter cells during cell division is fundamental to life. Segregation is mediated by interactions between dynamic microtubules and kinetochores, the megadalton protein complexes that assemble on the centromeres of each chromosome (Monda and Cheeseman, 2018; Musacchio and Desai, 2017). For accurate segregation, the sister kinetochores on each pair of chromatids must make bioriented attachments to microtubules emanating from opposite spindle poles. When linked sister kinetochores achieve biorientation, they come under tension due to pulling forces exerted by the opposing microtubules. However, because kinetochore–microtubule attachments initially form at random, erroneous connections lacking tension are often made. These must be detected and corrected to avoid missegregation. Tension appears to help cells distinguish correct from incorrect attachments, because attachments under tension are more stable than those lacking tension in vivo and in vitro (Akiyoshi et al., 2010; King and Nicklas, 2000; Nicklas and Koch, 1969).

A variety of error correction mechanisms help cells to make proper bioriented attachments. The most well-studied mechanism involves the conserved essential protein kinase Aurora B

(Krenn and Musacchio, 2015). When a kinetochore attaches incorrectly, the lack of tension is believed to signal Aurora B kinase to phosphorylate kinetochore proteins, which weakens their grip on the microtubule, causing detachment and giving the cell another chance to make a proper attachment (Biggins et al., 1999; Dewar et al., 2004; Hauf et al., 2003; Tanaka et al., 2002). When Aurora B is defective, cells are unable to make bioriented attachments (Biggins et al., 1999; Hauf et al., 2003; Tanaka et al., 2002). One of the major Aurora B substrates is Ndc80, a core component of the kinetochore that makes a significant contribution to kinetochore–microtubule coupling (Cheeseman et al., 2006; DeLuca et al., 2006). Ndc80 has two domains that mediate its interaction with the microtubule, a conserved calponin-homology "head" domain and a disordered N-terminal "tail" (Ciferri et al., 2008; Wei et al., 2007). The Ndc80 N-terminal tail contains multiple Aurora B consensus sites that underlie the regulation of Ndc80 by Aurora B (Akiyoshi et al., 2009; Cheeseman et al., 2006; Ciferri et al., 2008; DeLuca et al., 2006; Guimaraes et al., 2008; Miller et al., 2008; Wei et al., 2007). Additional kinetochore proteins are also phosphorylated by Aurora B to destabilize kinetochore–microtubule attachments, but

[1]Department of Physiology & Biophysics, University of Washington, Seattle, WA; [2]Howard Hughes Medical Institute, Chevy Chase, MD; [3]Division of Basic Sciences, Fred Hutchinson Cancer Research Center, Seattle, WA; [4]Molecular and Cellular Biology Program, University of Washington, Seattle, WA.

*K.K. Sarangapani, L.B. Koch, and C.R. Nelson contributed equally to this paper; Correspondence to Charles L. Asbury: casbury@uw.edu; Sue Biggins: sbiggins@fredhutch.org.

they vary depending on the organism (Cheeseman et al., 2002; Lan et al., 2004; Wordeman et al., 2007).

The Mps1 kinase is another conserved essential kinase implicated in kinetochore biorientation and error correction, independent of its well-studied role in signaling the spindle assembly checkpoint (Hewitt et al., 2010; Jelluma et al., 2008; Jones et al., 2005; Maciejowski et al., 2010; Maure et al., 2007; Santaguida et al., 2010). While there is some evidence that Mps1 regulates Aurora B activity (Jelluma et al., 2010; Saurin et al., 2011; Tighe et al., 2008), significant data suggest that Mps1 participates in error correction independently of Aurora B (Benzi et al., 2020; Hewitt et al., 2010; Maciejowski et al., 2010; Maure et al., 2007; Meyer et al., 2013; Santaguida et al., 2010). Consistent with this interpretation, inhibition of Mps1 does not alter the phosphorylation of many Aurora B substrates or Aurora B localization (Hewitt et al., 2010; Maciejowski et al., 2010; Maure et al., 2007; Santaguida et al., 2010; Tighe et al., 2008). Mps1 localizes to kinetochores by binding to the Ndc80 protein, and its activity is highest on kinetochores that have not made proper attachments (Hiruma et al., 2015; Ji et al., 2015; Kemmler et al., 2009; Kuijt et al., 2020). However, its role in regulating kinetochore–microtubule attachments and error correction has not been fully elucidated. In human cells, Mps1 regulates localization of the motor protein centromere protein E and the Ska complex, which stabilize proper kinetochore–microtubule attachments (Espeut et al., 2008; Hewitt et al., 2010; Maciejowski et al., 2017; Stucke et al., 2004). In budding yeast, Mps1 is required for localization of the Dam1 complex (Meyer et al., 2018; Meyer et al., 2013), an orthologue of the Ska complex (van Hooff et al., 2017). It also phosphorylates the Ndc80 protein, but this phosphorylation was reportedly involved in spindle checkpoint signaling and not in error correction (Kemmler et al., 2009). Recently, it was reported that Mps1 regulates biorientation via phosphorylation of the Spc105 protein (Benzi et al., 2020). While Mps1 targets many key microtubule-binding kinetochore elements, a unified view of how it participates in error correction remains elusive, in part because it is challenging to disentangle this function from the similar function of Aurora B and the well-established role for Mps1 in checkpoint signaling.

To directly study the effect of Mps1 activity on kinetochore–microtubule attachments, we took advantage of a yeast reconstitution system where the strength of attachment between individual isolated kinetochores and single microtubules can be measured in vitro (Akiyoshi et al., 2010; Sarangapani et al., 2013). Using isolated kinetochore particles that copurify with Mps1 kinase but lack other kinase activity (London et al., 2012), we found that Mps1 phosphorylation of Ndc80 directly weakens kinetochore–microtubule attachments, similar to the function of Aurora B. Moreover, phosphorylation of Ndc80 in vivo by Mps1 occurs during mitosis, when kinetochores are prevented from coming under tension. Cells containing mutations in the Mps1-targeted phosphorylation sites on Ndc80 exhibit genetic interactions and chromosome segregation defects when combined with inhibitory mutations in other error correction pathways (Akiyoshi et al., 2010; Cheeseman et al., 2002; Miller et al., 2019; Miller et al., 2016). Taken together, our data suggest that Mps1-mediated phosphorylation of Ndc80 provides an additional mechanism for

specifically weakening kinetochore–microtubule attachments that lack tension, complementing the Aurora B and intrinsic error correction pathways, thereby helping to release erroneous attachments and ensure the accuracy of chromosome segregation.

## Results

### Copurifying kinase activity weakens the attachment of isolated kinetochores to microtubules

We previously found that native kinetochore particles purified from budding yeast lack Aurora B kinase activity and that the major copurifying kinase activity is due to Mps1 (London et al., 2012). To test whether the copurifying Mps1 activity directly affects interactions between kinetochores and microtubules, we modified our previously developed approach for measuring the strengths of individual kinetochore–microtubule attachments in vitro (Akiyoshi et al., 2010), by adding ATP to activate any copurifying kinase. We isolated kinetochores via anti-Flag immunoprecipitation of the Dsn1 protein (Dsn1-6His-3Flag; Fig. S1 A). After elution with Flag peptide, we linked the native kinetochore particles to polystyrene microbeads, mixed them with ATP to activate the copurifying kinase, and immediately introduced them into a flow chamber containing dynamic microtubules grown from coverslip-anchored microtubule seeds (Fig. 1 A). During the time required to seal the slide chamber and mount it onto the laser trap (~10 min), some kinetochore-decorated beads attached spontaneously to the sides of coverslip-anchored microtubules. The laser trap was used to bring these beads that were laterally attached to microtubules to the plus end tips and then apply gradually increasing force until the kinetochores ruptured from the tips (Fig. 1, B and C). Rupture strengths for many individual attachments were collected in the presence of ATP, ADP, or without adenosine until 90 min after sealing the slide. For each population, the median rupture force was calculated, and the fraction of attachments that survived up to a given level of force was plotted. As in our previous work (Akiyoshi et al., 2010; Sarangapani et al., 2013), kinetochore particles not exposed to ATP ruptured over a range of forces, with a median strength of 9.8 pN in the absence of adenosine, or 8.7 pN in the presence of ADP (Fig. 1 D). However, when ATP was included, the rupture force distribution was shifted to lower values, and the median strength was only 5.3 pN, suggesting that phosphorylation by a kinetochore-associated kinase decreased the strength of kinetochore–microtubule attachments. Consistent with this interpretation, if λ-phosphatase was included together with the ATP, then the ATP-dependent weakening was reduced, and the kinetochores maintained a median strength of 7.5 pN, similar to untreated and ADP-treated kinetochores (Figs. 1 D and S1 B). We monitored the rupture strength as a function of time, which indicated that the ATP-dependent weakening reaction was completed during the ~10-min slide preparation, with no significant weakening thereafter (Fig. S1 C). We also confirmed that ATP treatment did not alter the core composition of the kinetochore particles and that Mps1 was released due to autophosphorylation as previously reported (Fig. S1 D; Koch et al., 2019). Altogether, these data indicate that phosphorylation of native kinetochores by a copurifying kinase activity reduces the strength of their attachments to microtubules.

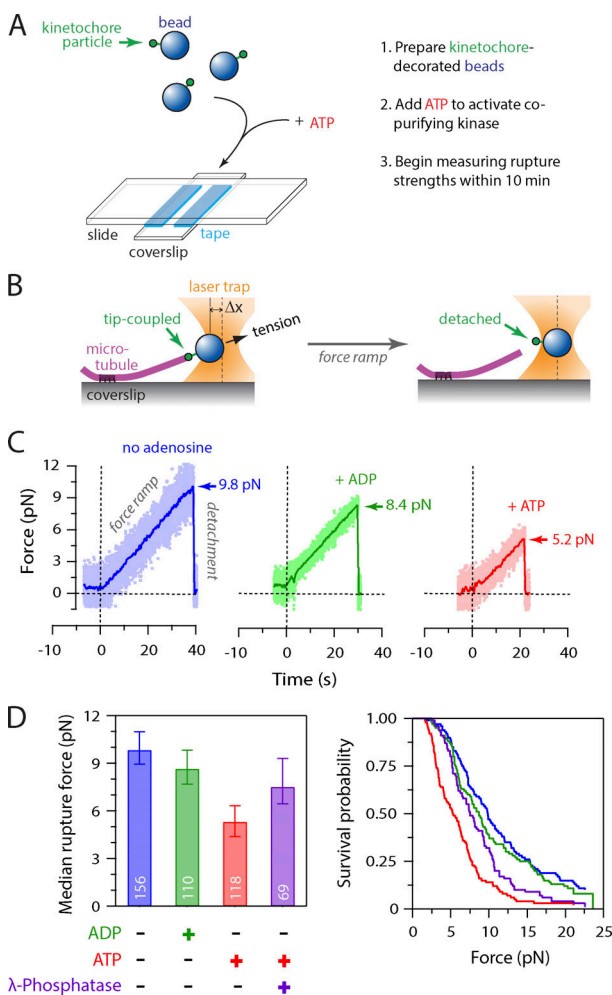

Figure 1. **Activity of copurifying kinase weakens attachment of native kinetochores to microtubules. (A)** Schematic of slide preparation. Kinetochore-decorated beads were prepared; mixed with ATP (or ADP as a control), free tubulin, and GTP; and introduced into a flow chamber containing coverslip-anchored microtubule seeds. **(B)** Schematic of laser trap assay. After a kinetochore-decorated bead was tip-coupled under low preload (left), the force was gradually increased until detachment (right). Trap measurements commenced ~10 min after mixing kinetochore-decorated beads with ATP (or ADP) and continued for up to 90 min. **(C)** Example traces showing forced detachment under indicated conditions of beads decorated with native kinetochore particles isolated from WT cells (SBY8253). Dots represent instantaneous force fluctuations. Solid traces show same data after smoothing with a 500-ms sliding boxcar average. Dashed vertical lines mark the start of the 0.25 pN • s⁻¹ force ramp. Arrows mark rupture force. **(D)** Median rupture strengths (left) and corresponding survival probability distributions (right) for WT kinetochores (from SBY8253) under indicated conditions. ATP exposure (red) caused significant weakening, which was inhibited by addition of λ-phosphatase (purple). Values inside bars indicate numbers of events for each condition. Error bars represent ± 95% confidence intervals calculated by bootstrapping. P values for all pairwise strength comparisons (from log-rank tests) are provided in Table S5.

## Mps1 activity is required for the ATP-dependent weakening of isolated kinetochores

Because Mps1 was the only kinase activity we detected on purified kinetochores (London et al., 2012), we next tested whether Mps1 mediates the ATP-dependent weakening of kinetochore particles using an Mps1 inhibitor called reversine (Santaguida

et al., 2010). We first verified that reversine inhibits Mps1 activity on the particles by incubating them with radioactive ATP in the presence or absence of the drug. Autoradiography revealed phosphorylation on Spc105, Ndc80, and Dsn1 when reversine was omitted (Fig. 2 A), in agreement with our previous work (London et al., 2012). When reversine was added, total phosphorylation levels on all the substrates decreased by 69%, confirming significant inhibition. We then performed rupture force assays, mixing kinetochore-decorated microbeads with reversine plus ATP or with the vehicle DMSO plus ATP, ADP, or no adenosine as controls. The median strength of kinetochores measured in the presence of DMSO was 8.2 pN without adenosine and fell to 5.0 pN in the presence of ATP. This loss of

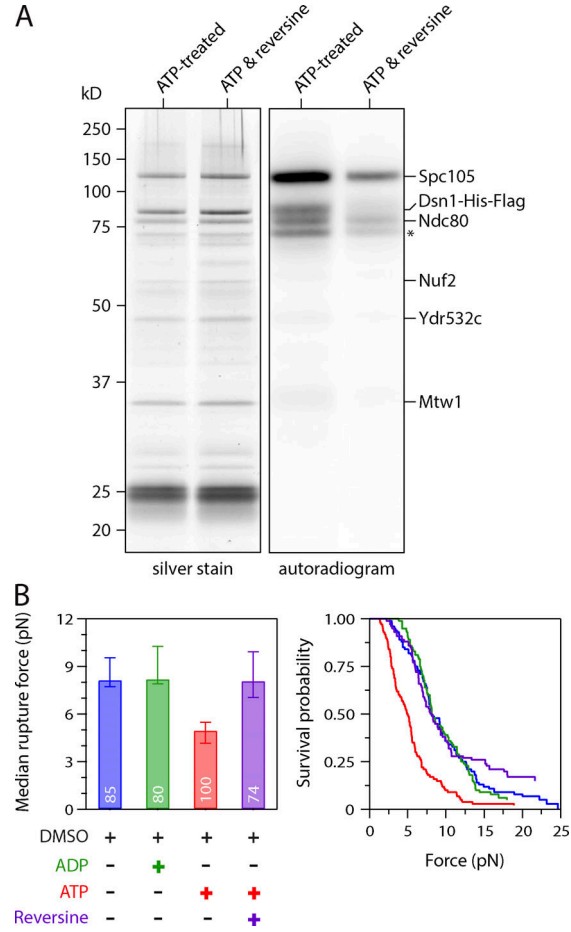

Figure 2. **ATP-dependent weakening of native kinetochores requires Mps1 kinase activity. (A)** The Mps1 inhibitor reversine reduces phosphorylation of Spc105, Dsn1, and Ndc80 by copurifying kinase activity. Kinetochores purified by immunoprecipitation of Dsn1-His-Flag from WT cells (SBY8253) were incubated with γ-³²P-labeled ATP, either alone or with 5 µM of reversine, and then visualized by silver stain or autoradiography after SDS-PAGE. Asterisk marks an additional unidentified phosphoprotein. **(B)** Median rupture strengths (left) and corresponding survival probability distributions (right) for WT kinetochores (from SBY8253) measured under indicated conditions. The ATP-dependent weakening (red) was blocked by addition of 5 µM reversine (purple). Values inside bars indicate numbers of events for each condition. Error bars represent ± 95% confidence intervals calculated by bootstrapping. P values for all pairwise strength comparisons (from log-rank tests) are provided in Table S5.

strength was completely blocked by reversine, suggesting it was due to Mps1 (Fig. 2 B). To further confirm that Mps1 activity was required, we purified kinetochores from *mps1-1* mutant cells and performed rupture force assays. The ATP-dependent weakening of kinetochore particles isolated from *mps1-1* cells was also suppressed (Fig. S2 A). In addition, although Mps1 recruits Bub1, very little Bub1 kinase copurifies with the kinetochore particles, and they carry no detectable Bub1 kinase activity (Akiyoshi et al., 2010; London and Biggins, 2014a). Taken together, these data strongly suggest that Mps1 is the key kinase that causes the ATP-dependent decrease in kinetochore strength.

## ATP-dependent weakening depends on phosphorylation of Ndc80 not Spc105

We next sought to identify the key substrates whose phosphorylation by Mps1 causes the weakening of reconstituted kinetochore–microtubule attachments. Mps1-mediated phosphorylation of Spc105 is vital for initiating spindle checkpoint signaling (London et al., 2012; Shepperd et al., 2012; Yamagishi et al., 2012), and recent work suggests it also promotes kinetochore biorientation through recruitment of the Bub1 protein (Benzi et al., 2020; Storchová et al., 2011). To test whether Spc105 is the relevant substrate underlying ATP-dependent kinetochore weakening in vitro, we purified kinetochores from phospho-deficient *spc105-6A* cells, which carry alanine substitutions to block phosphorylation at all six Mps1 phosphorylation sites (the MELT motifs) within the disordered N-terminal region of Spc105 (Fig. 3 A; London et al., 2012). The composition of the kinetochores purified from *spc105-6A* cells was similar to that of WT, as assayed by silver-stained PAGE and immunoblotting against representative kinetochore proteins (Fig. 3, B and C). In rupture force assays, the median strength of phospho-deficient Spc105-6A kinetochores was 8.3 pN without adenosine and fell to 3.2 pN in the presence of ATP (Fig. 3 D). We were surprised that the Spc105-6A kinetochores were weaker after ATP addition than their WT counterparts. While the reason for this enhanced effect of ATP on Spc105-6A kinetochores is unclear, the rupture strength data nevertheless indicate that Spc105 is not the key Mps1 target underlying ATP-dependent weakening of the kinetochores in vitro.

We next focused on the major microtubule binding component of the kinetochore, Ndc80c (Akiyoshi et al., 2010; Cheeseman et al., 2006; DeLuca et al., 2006). Previous work identified a total of 14 sites on the Ndc80 protein that are phosphorylated by Mps1 (Fig. 3 E; Kemmler et al., 2009). However, three of these overlap with known Aurora B phosphosites (Akiyoshi et al., 2009), so we tested whether the remaining 11 are involved in the ATP-dependent weakening. We purified kinetochore particles from phospho-deficient *ndc80-11A* cells, which carry alanine substitutions at all 11 of the Mps1 phosphorylation sites, to prevent their phosphorylation (Fig. S2 B; Kemmler et al., 2009). Similar levels of Mps1 copurified with both Ndc80-11A and WT kinetochores (Fig. S2 C), but the kinase activity had no effect on the rupture strengths of Ndc80-11A kinetochores. The median strength of Ndc80-11A kinetochores was 8.7 pN in the absence of adenosine, similar to WT

kinetochores (Fig. 3 F). Upon exposure to ATP, the median strength of Ndc80-11A kinetochores remained high, 8.5 pN, which is indistinguishable from the strength measured without adenosine (Figs. 3 F and S2 D). These data indicate that one or more of the Mps1 phosphorylation sites on Ndc80 are required for the decreased attachment strength.

## Mps1 phosphorylation of Ndc80 is not required for spindle assembly checkpoint signaling

A previous report suggested that Mps1-mediated phosphorylation of Ndc80 regulates the spindle assembly checkpoint and does not affect kinetochore–microtubule attachments or chromosome segregation (Kemmler et al., 2009), a conclusion that seemed inconsistent with our in vitro results. We therefore reexamined the phenotypes of mutant *ndc80-14A* cells, with alanine substitutions at all 14 Mps1 target sites included for consistency with the earlier study. To test whether *ndc80-14A* mutants are defective in the spindle assembly checkpoint, we released WT and *ndc80-14A* cells from G1 into the microtubule depolymerizing drug nocodazole, which generates unattached kinetochores that normally trigger the spindle assembly checkpoint, and we monitored cell cycle progression by analyzing levels of the anaphase inhibitor Pds1/securin (Cohen-Fix et al., 1996). Pds1 levels accumulated and then stabilized in both WT and *ndc80-14A* cells as they progressed through the cell cycle and then arrested (Fig. 4 A). The stabilization of Pds1 in both WT and *ndc80-14A* cells depended on the spindle assembly checkpoint, because it was eliminated in both strains by deletion of the checkpoint gene *MAD2* (Fig. 4 A). These observations indicate that blocking the phosphorylation of all known Mps1 target sites on the Ndc80 protein does not lead to a defective spindle assembly checkpoint as previously reported (Kemmler et al., 2009). It was also reported that replacing all 14 Mps1 target residues with phospho-mimetic aspartic acid residues was lethal due to constitutive activation of the spindle assembly checkpoint (Kemmler et al., 2009). To test this, we performed a plasmid shuffle assay using strains expressing a WT copy of *NDC80* from a plasmid that also contains the *URA3* gene, which renders cells susceptible to the cytotoxic drug 5-fluoroorotic acid (5-FOA; Boeke et al., 1987). Cells carrying a functional chromosomal copy of *NDC80* can spontaneously lose the *NDC80-URA3* plasmid and are therefore able to grow on medium containing 5-FOA, whereas cells with a nonfunctional chromosomal allele require the plasmid for viability and are killed by 5-FOA. Using this assay, we confirmed that *ndc80-14D* is recessive lethal (Fig. 4 B). However, in contrast to the previous report (Kemmler et al., 2009), deleting the spindle assembly checkpoint gene *MAD2* did not rescue this lethality (Fig. 4 B). Notably, spurious colonies appeared after the *ndc80-14D* mutants were grown for longer times, suggesting the existence of spontaneous suppressor mutations (Fig. S3). While our data indicate that mutation of the previously identified Mps1 phosphorylation sites on Ndc80 does not cause defects in spindle assembly checkpoint signaling, they confirm the previously reported lethality of *ndc80-14D* (Kemmler et al., 2009). Therefore, one or more of the Mps1 phosphorylation sites have a critical cellular function.

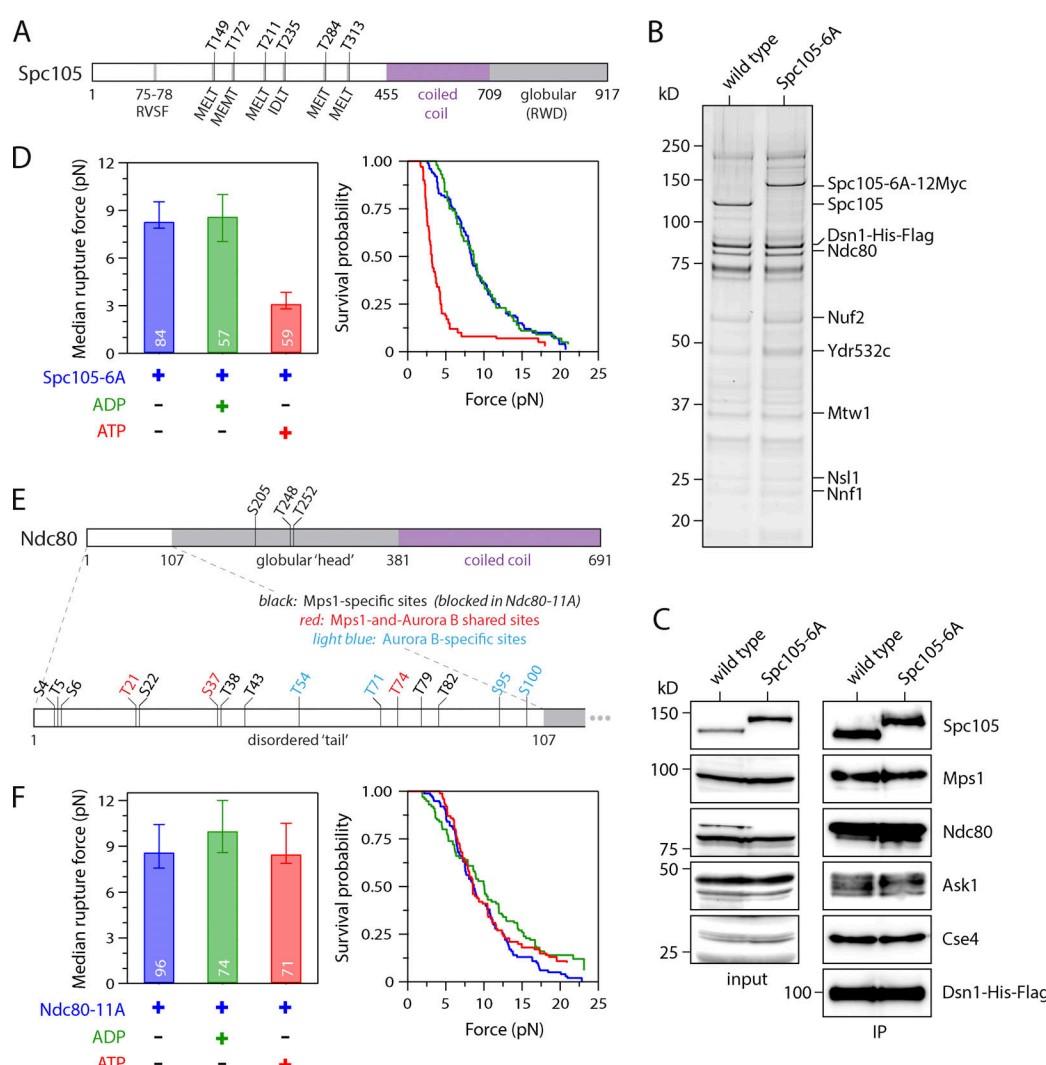

Figure 3. **ATP-dependent weakening depends on phosphorylation of the Ndc80 tail, not of Spc105. (A)** Map of six Mps1 phosphorylation target sites (MELT motifs) within the disordered N-terminus of Spc105. **(B)** Kinetochores purified from mutant cells carrying phospho-deficient Spc105-6A (SBY10315) and from WT cells (SBY8253) visualized by silver stain after SDS-PAGE. **(C)** Kinetochore material purified as in B and then analyzed by immunoblotting for the indicated kinetochore components. No Dsn1-His-Flag blot is shown for the input material because Dsn1 was undetectable in whole-cell lysates, due to an overlapping cross-reactive band. IP, immunoprecipitation. **(D)** Median rupture strengths (left) and corresponding survival probability distributions (right) for phospho-deficient Spc105-6A kinetochores (from SBY10315). These kinetochores were weakened upon exposure to ATP by an amount similar to kinetochores carrying WT Spc105. **(E)** Map of Mps1 and Aurora B phosphorylation target sites within the globular head (top) and disordered N-terminal tail (bottom) of Ndc80. The tail includes three shared sites targeted by both Mps1 and Aurora B (shown in red), plus eight sites targeted specifically by Mps1 (black). (Four additional sites targeted specifically by Aurora B are shown in light blue.) **(F)** Median rupture strengths (left) and corresponding survival probability distributions (right) for phospho-deficient Ndc80-11A kinetochores (from SBY19838), with Ala substitutions blocking all 11 of the Mps1-specific target sites (i.e., those colored black in E), measured under indicated conditions. These kinetochores were refractory to ATP-dependent weakening. Residue numbers below the maps in A and E demarcate major sequence features. Values inside bars in D and F indicate numbers of events for each condition. Error bars in D and F represent ± 95% confidence intervals calculated by bootstrapping. P values for all pairwise strength comparisons (from log-rank tests) are provided in Table S5.

## Weakening of kinetochore–microtubule attachments occurs via phosphorylation of Mps1-specific targets in the N-terminal tail of Ndc80

Our observation that Mps1-mediated phosphorylation of Ndc80 weakens reconstituted kinetochore–microtubule attachments is similar to the well-documented function of Aurora B–mediated phosphorylation of Ndc80, which inhibits kinetochore–microtubule attachments in vivo and in vitro (Cheeseman et al., 2006; Ciferri et al., 2008; DeLuca et al., 2006; Sarangapani et al.,

2013; Wei et al., 2007). Eight of the 11 Mps1-specific target sites on Ndc80 fall within the disordered N-terminal tail domain, which is also where the key Aurora B target sites are located (Fig. 3 E). To analyze the contribution of the Mps1-specific sites within the Ndc80 tail (Fig. 5 A), we generated a mutant with alanine substitutions at just these eight sites (Nc80-8A). We initially included an epitope tag (–3HA) to allow direct immunoprecipitation of the phospho-deficient Ndc80-8A, or of WT Ndc80 as a control. Mps1 copurified with WT Ndc80, as

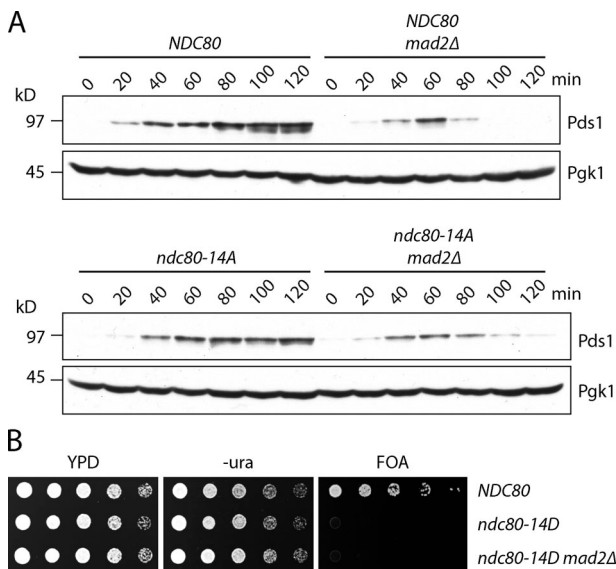

Figure 4. **Mps1 phosphorylation of Ndc80 is not required for spindle assembly checkpoint signaling. (A)** Cells carrying phospho-deficient *ndc80-14A* maintain a spindle assembly checkpoint arrest in response to nocodazole. The indicated strains (SBY17648, SBY17624, SBY17807, and SBY17895) were initially arrested in G1 using α-factor for 3 h. α-Factor was then washed out, and the cells were resuspended in medium with 10 μg • ml⁻¹ nocodazole. Samples taken at indicated time points were analyzed for the anaphase inhibitor Pds1/securin (and Pgk1 as a loading control) by immunoblotting. **(B)** The phospho-mimetic *ndc80-14D* allele is recessive lethal, and its lethality is not rescued by deletion of the checkpoint gene *MAD2*. The indicated strains (SBY15149, SBY9453, and SBY15087) were tested using a plasmid shuffle assay, where growth on 5-FOA plates indicates viability in the absence of a covering copy of WT *NDC80*. Fivefold serial dilutions were plated onto YPD, –uracil, or 5-FOA plates and grown at 23°C.

previously demonstrated (Kemmler et al., 2009), and we found that a similar level of Mps1 copurified with Ndc80-8A (Fig. 5 B). After incubating both immunoprecipitations with radioactive ATP, autoradiography showed 49% less phosphorylation on Ndc80-8A relative to WT (Fig. 5 B), confirming that Mps1 phosphorylates Ndc80 at one or more of the eight Mps1-specific target sites within its N-terminal tail domain.

To test whether these Mps1-specific sites in the Ndc80 tail are involved in ATP-dependent weakening of kinetochore–microtubule attachments, we isolated kinetochores from strains containing WT or Ndc80-8A protein (via anti-Flag-based immunoprecipitation of Dsn1-6His-3Flag). Based on silver staining after SDS-PAGE and mass spectrometry analysis, we verified that the composition of the purified Ndc80-8A kinetochores was similar to WT (Fig. S4). In the rupture force assay, the Ndc80-8A kinetochores had a median strength of 9.4 pN when no adenosine was included, and a strength of 8.6 pN in the presence of ADP, values similar to WT kinetochores (Fig. 5 C). However, unlike WT kinetochores, the Ndc80-8A kinetochores were unaffected by exposure to ATP, maintaining a high median rupture strength of 9.4 pN (Fig. 5 C). To further test the importance of the Mps1-specific tail sites, we also purified Ndc80-8D kinetochores carrying aspartic acid substitutions at all eight sites, to mimic their phosphorylation. The phospho-mimetic Ndc80-8D kinetochores had a composition similar to WT kinetochores (Fig.

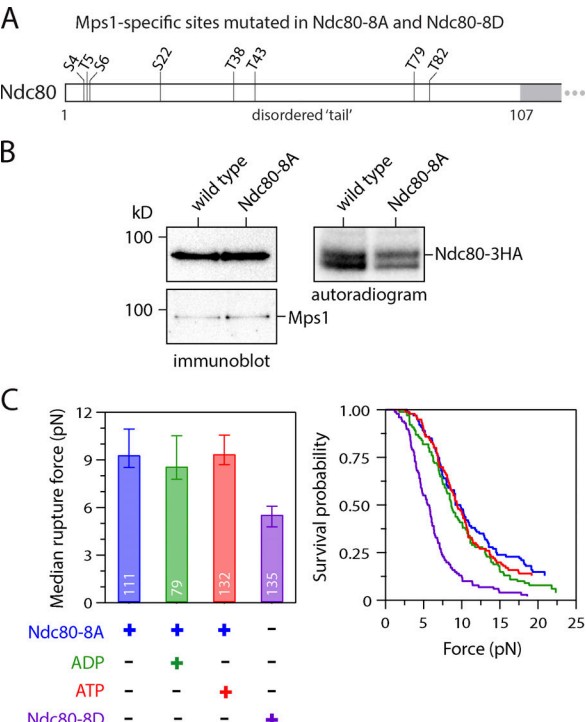

Figure 5. **ATP-dependent weakening depends on phosphorylation of Mps1-specific target sites in the Ndc80 tail. (A)** Map of Mps1-specific target sites within the N-terminal tail of Ndc80. All eight sites were mutated to Ala in phospho-deficient Ndc80-8A and to Asp in phospho-mimetic Ndc80-8D. **(B)** In vitro phosphorylation of Ndc80-8A was reduced relative to WT Ndc80, even though similar levels of Mps1 were copurified. WT Ndc80 and phospho-deficient Ndc80-8A were purified by immunoprecipitation (from SBY20062 and SBY20063), incubated with ³²P-γ-labeled ATP, and visualized by immunoblotting (left) or autoradiography (right) after SDS-PAGE. **(C)** Median rupture strengths (left) and corresponding survival probability distributions (right) for phospho-deficient Ndc80-8A and phospho-mimetic Ndc80-8D kinetochores (purified from SBY19855 and SBY19877, respectively) measured under indicated conditions. The Ndc80-8A kinetochores were refractory to ATP-dependent weakening (red), whereas Ndc80-8D kinetochores were constitutively weak (purple). Values inside bars indicate numbers of events for each condition. Error bars represent ± 95% confidence intervals calculated by bootstrapping. P values for all pairwise strength comparisons (from log-rank tests) are provided in Table S5.

S4), but their rupture strength was constitutively low, with a median rupture strength of only 5.6 pN in the absence of adenosine (Fig. 5 C). Altogether, these observations suggest that Mps1 phosphorylation of the Ndc80 tail is necessary and sufficient for directly weakening the isolated kinetochores upon exposure to ATP.

### Phosphorylation of Ndc80 by Mps1 in vivo is stimulated when kinetochores lack tension

To determine whether Mps1 phosphorylates Ndc80 in vivo, we set out to generate an antibody that recognizes an Mps1 site. Although we were not able to produce one against a site specific to Mps1 phosphorylation, we succeeded in generating a polyclonal antibody that recognizes Ndc80 carrying a phosphate modification at Thr-74 (pT74), a site that is phosphorylated by both Ipl1 and Mps1 (Akiyoshi et al., 2009; Kemmler et al., 2009).

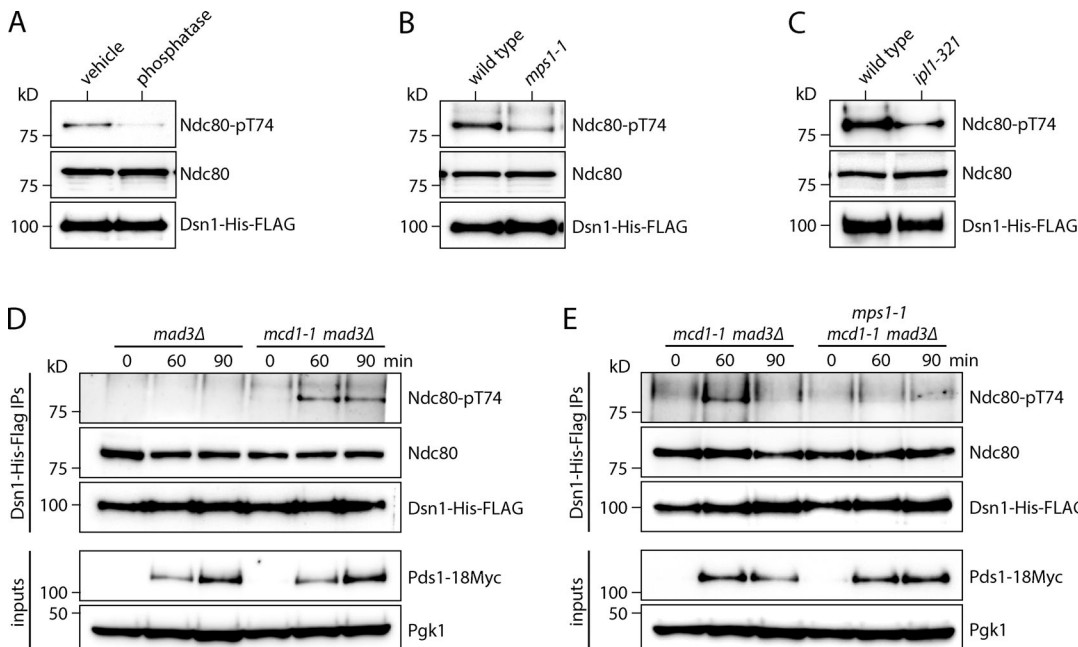

Figure 6. **Phosphorylation of Ndc80 by Mps1 in vivo is stimulated when kinetochores lack tension. (A)** Kinetochores purified from WT cells (SBY8253) were treated with or without λ-phosphatase for 30 min at 30°C. Phosphorylation of Ndc80 at Thr-74 (Ndc80-pT74) was visualized by immunoblotting. **(B)** Kinetochores were purified from WT (SBY8253) or *mps1-1* (SBY8726) cells after a 3-h 37°C temperature shift. Phosphorylation of Ndc80 at Thr-74 was visualized by immunoblotting. **(C)** Kinetochores were purified from WT (SBY8253) or *ipl1-321* (SBY8721) cells after a 3-h 37°C temperature shift. Phosphorylation of Ndc80 at Thr-74 was visualized by immunoblotting. **(D)** Phosphorylation of Ndc80 on kinetochores at Thr-74 was analyzed 30, 60, and 90 min after G1 release in *mad3Δ* (SBY20362) and *mad3Δ mcd1-1* (SBY20361) cells via immunoblotting. Cells were arrested in α-factor and released at 37°C, and kinetochore purifications were performed at indicated time points. Pds1-18Myc immunoblotting was used to indicate mitotic timing. **(E)** Phosphorylation of Ndc80 on kinetochores at Thr-74 was analyzed (as in D) in *mad3Δ mcd1-1* (SBY20361) and *mad3Δ mcd1-1 mps1-1* (SBY20622) cells.

On immunoblots of purified kinetochore complexes, this Ndc80-pT74 antibody detected a signal corresponding to the Ndc80 protein that was diminished when the kinetochores were treated with λ-phosphatase (Fig. 6 A), confirming the phospho-specificity of the antibody. To check whether Mps1 contributes to the phosphorylation of this site, we purified kinetochores from mutant *mps1-1* cells, in which Mps1 is specifically in-activated when the cells are grown at a nonpermissive temperature (Fig. 6 B). The Ndc80 signal detected in WT kinetochore particles by the Ndc80-pT74 antibody was reduced when mutant *mps1-1* kinetochores were probed. To confirm that Aurora B also phosphorylates this residue, we purified kinetochores from *ipl1-321* cells that lack Aurora kinase activity and also found a reduction in phosphorylation of T74 on Ndc80 (Fig. 6 C; Biggins et al., 1999). Taken together, these data show that both Mps1 and Aurora B contribute to Ndc80-T74 phosphorylation on kinetochores in vivo.

A lack of tension on erroneous kinetochore–microtubule at-tachments is thought to promote their release by triggering Aurora B–mediated kinetochore phosphorylation (Biggins and Murray, 2001; Nicklas and Koch, 1969; Stern and Murray, 2001; Tanaka et al., 2002). To test whether a lack of tension can also trigger Mps1-mediated kinetochore phosphorylation in vivo, we analyzed the phosphorylation of Ndc80-T74 on ki-netochores in a mutant *mcd1-1* strain, which cannot generate tension on its kinetochores due to a defect in sister chromatid cohesion (Stern and Murray, 2001; Tanaka et al., 2000). WT and

*mcd1-1* mutant cultures were released from G1 to the nonper-missive temperature, and kinetochores were purified at various time points. The cells had *MAD3* deleted to ensure equivalent progression through the cell cycle, which we verified by ana-lyzing Pds1 levels in the lysates (Fig. 6 D). Immunoblotting confirmed that kinetochores purified from either WT or *mcd1-1* cells contained similar amounts of Ndc80 protein. In the *mad3Δ* background, we detected weak Ndc80-T74 phosphorylation on WT kinetochores at time points corresponding to mitosis. However, kinetochores purified from the cohesion-deficient *mcd1-1* cells showed an enrichment of phosphorylation at Ndc80-T74 (Fig. 6 D), indicating that this phosphorylation is enhanced in vivo under conditions where kinetochores lack tension. To test whether Mps1 contributes to this phosphoryl-ation, we repeated the experiment and compared *mcd1-1 mad3Δ* cells to *mcd1-1 mad3Δ mps1-1* cells. Strikingly, the phosphorylation that occurs in the cohesion mutant was reduced (but not elimi-nated) in the absence of Mps1 activity (Fig. 6 E), suggesting that Mps1 contributes to Ndc80 phosphorylation in vivo in response to tension defects.

## Mps1 phosphorylation of Ndc80 contributes to error correction in vivo

Aurora B phosphorylation of Ndc80 promotes error correction, so we reasoned that Mps1 phosphorylation of Ndc80 might play a similar role in vivo. To test this hypothesis, we treated WT and *ndc80-8A* cells with nocodazole, to depolymerize their spindles

and detach their kinetochores, and then washed out the noco-dazole to allow spindle reformation, creating a situation where the cells had to correct erroneous attachments to achieve proper kinetochore biorientation. We measured the time required for the cells to make proper bioriented attachments by analyzing the separation of fluorescently marked centromere 8, which splits into two foci upon kinetochore biorientation. Bi-orientation in the *ndc80-8A* cells was delayed by ~15 min rel-ative to WT, consistent with Mps1 phosphorylation promoting error correction (Fig. 7 A). However, mutating the Aurora B and Mps1 phosphorylation sites in the Ndc80 tail does not affect cell growth (Fig. 7, B and C; Akiyoshi et al., 2009), consistent with the involvement of multiple pathways in error correction. One such pathway relies on Aurora B phosphorylation of the Dam1 complex (Cheeseman et al., 2002; Tien et al., 2010). We therefore reasoned that if Mps1 phosphorylation of Ndc80 provides another error correction pathway, then mutation of Aurora B target sites within the Dam1 complex might interact genetically with mutation of the Mps1 sites within Ndc80. In-deed, we found that the *ndc80-8D* mutant is synthetically lethal when combined with the *dam1-3D* mutant (Fig. 7 B). Another previously studied pathway that contributes to error correction is the direct stabilization of kinetochore–microtubule attach-ments by tension (Akiyoshi et al., 2010), an intrinsic property of kinetochores that requires kinetochore-bound Stu2 (Miller et al., 2019; Miller et al., 2016). Mutants in Aurora B exhibit genetic interactions with *stu2^ccΔ*, an allele of Stu2 that specifi-cally causes defects in kinetochore biorientation (Miller et al., 2019). We therefore reasoned that the Mps1-dependent error correction pathway might also exhibit growth defects in com-bination with *stu2^ccΔ*. Indeed, although the *ndc80-8A* cells car-rying phospho-blocking substitutions at Mps1-specific target sites are viable, we found that combining *ndc80-8A* with *stu2^ccΔ* exacerbated the growth defect of *stu2^ccΔ* alone (Fig. 7 C). To analyze chromosome segregation in these cells, we generated *ndc80-8A stu2^ccΔ* strains with a fluorescent marker on a single chromosome and with the endogenous WT *STU2* gene under control of a conditional auxin-inducible degron (*stu2-AID*) to maintain viability. Because *stu2^ccΔ* causes a spindle assembly checkpoint–dependent arrest, we also deleted the *MAD3* gene to enable the cells to progress to anaphase. After arresting the cells in G1, we released them into growth medium with auxin to repress the endogenous *stu2-AID* gene and then analyzed chromosome segregation in anaphase. Consistent with our prior work (Miller et al., 2019), chromosome missegregation was increased in *stu2^ccΔ mad3Δ* cells relative to *mad3Δ* control cells (6.5% in *STU2^WT mad3Δ* versus 19.4% in *stu2^ccΔ mad3Δ* cells; Fig. 7 D). The addition of *ndc80-8A* strongly exacerbated the chromosome missegregation defect (30.3%), indicating that Mps1 phosphorylation of Ndc80 contrib-utes to accurate chromosome segregation.

## Discussion

To study the effects of kinetochore-associated Mps1 on kinetochore–microtubule attachments independently of other cellular kinases or pathways, we devised a reconstitution ap-proach where the endogenous Mps1 that copurifies with isolated

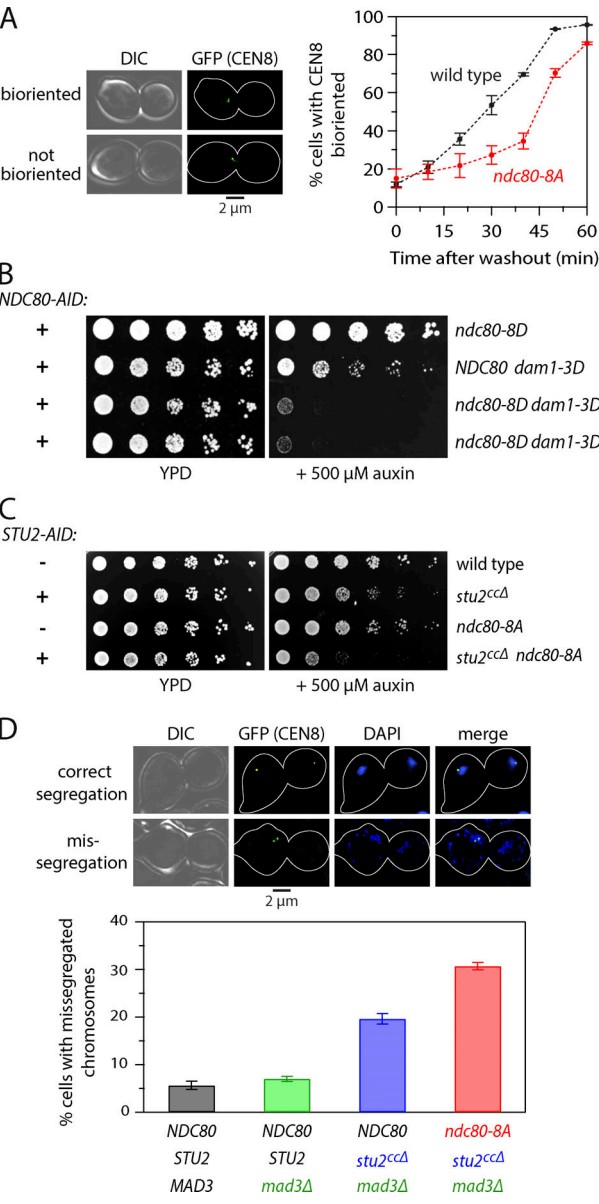

Figure 7. **Mps1 phosphorylation of Ndc80 contributes to error correc-tion in vivo. (A)** Analysis of biorientation. WT (SBY20894) and *ndc80-8A* (SBY20893) cells were arrested in metaphase with a *cdc20-AID* system for 3 h, with 10 µg • ml⁻¹ nocodazole added during the last 30 min. Nocodazole was then washed out, and biorientation was scored at the indicated times by examining GFP markers on centromere 8 (CEN8; example images at left). Mean percentages (± SEM) from three independent experiments where at least 200 cells were analyzed are plotted and the SE is reported (at right). **(B)** *ndc80-8D* (SBY20201), *dam1-3D* (SBY20199), and *ndc80-8D dam1-3D* (SBY20217) cells all expressing an *NDC80-AID* system were serially diluted and spotted onto YPD or YPD + 500 µM auxin plates at 23°C. **(C)** WT cells (SBY20062), *stu2-ccΔ* cells expressing a *STU2-AID* system (SBY20170), *ndc80-8A* cells (SBY20063), and *ndc80-8A stu2-ccΔ* cells expressing a *STU2-AID* system (SBY20171) were serially diluted and spotted on YPD or YPD + 500 µM auxin plates at 30°C. **(D)** Chromosome segregation analyzed in *STU2-AID* cells expressing *STU2-3V5* (SBY17527), *STU2-3V5 mad3Δ* (SBY17668), *stu2-ccΔ mad3Δ NDC80-3HA* (SBY20210), or *stu2-ccΔ mad3Δ ndc80-8A-3HA* (SBY20211). Segregation fidelity was measured by examining GFP markers on centromere 8 (CEN8; example images at top) and DNA distribution (DAPI stain) in large-budded cells 2 h after α-factor release. Mean percentages (± SEM) from three independent experiments where at least 200 cells were analyzed are plotted and the SE is reported. DIC, differential interference contrast.

kinetochores is activated via exposure to ATP immediately before laser trap strength measurements. Our data show that Mps1-mediated phosphorylation of the N-terminal tail of Ndc80 causes a direct weakening of the kinetochore–microtubule interface in vitro. This function was previously difficult to identify in vivo due to the presence of Aurora B, which can also phosphorylate Ndc80. Our in vitro approach allowed us to separate their functions. We further show that Mps1 phosphorylation of Ndc80 occurs in vivo and is enhanced when kinetochores lack tension due to defective sister chromatid cohesion. Moreover, specifically blocking this Mps1-mediated phosphorylation exacerbates chromosome missegregation when combined with a *stu2* mutant defective in another kinase-independent, intrinsic error correction pathway (Miller et al., 2019; Miller et al., 2016). Together, these observations suggest that Mps1 contributes directly to the release of erroneous kinetochore–microtubule attachments, consistent with the previous observation that it is required for the reorientation of kinetochores lacking tension in vivo (Maure et al., 2007).

A previous report identified a total of 14 Mps1 target sites on Ndc80 and suggested that their phosphorylation was important for regulating the spindle assembly checkpoint without affecting kinetochore–microtubule attachments (Kemmler et al., 2009). While we confirmed the lethality of the phospho-mimetic *ndc80-14D* mutant, its lethality was not suppressed in our hands when the checkpoint was eliminated, nor was our phospho-deficient *ndc80-14A* strain defective in checkpoint signaling as reported (Kemmler et al., 2009). These differences in our findings may be due to different strain backgrounds or the possibility that suppressors can arise more easily in the absence of the checkpoint. The *ndc80-8D* strain with phospho-mimetic substitutions at only the eight Mps1-specific target sites within the Ndc80 tail produced constitutively weak kinetochores in vitro but nevertheless remained viable (Sarangapani et al., 2013). Its viability indicates that the lethality of *ndc80-14D* arises from its additional substitutions, including those at Mps1 sites outside the N-terminal tail, whose function will be important to identify in the future.

Phosphorylation at one or more of the eight Mps1-specific target sites within the Ndc80 tail is necessary and sufficient for Mps1-mediated weakening of reconstituted kinetochore–microtubule attachments. While other yeast kinetochore proteins such as Spc105 have been identified as substrates of Mps1 (Benzi et al., 2020; London and Biggins, 2014a; London and Biggins, 2014b; Shimogawa et al., 2006), Spc105 phosphorylation does not appear to directly regulate kinetochore attachment strength. The Ndc80 tail is also a major target of regulation by Aurora B kinase, which in yeast phosphorylates four distinct Aurora B–specific phospho-sites, plus three shared sites phosphorylated by both Mps1 and Aurora B. Phosphorylation of the human Ndc80 (Hec1) tail by Cdk1 kinase was also identified recently and was implicated in the correction of erroneous kinetochore–microtubule attachments (Kucharski et al., 2021). Aurora A also phosphorylates a site in the Ndc80 tail during mitosis (DeLuca et al., 2018; Ye et al., 2015). Why multiple different kinases phosphorylate a single domain within Ndc80 is not yet clear, but this convergence might allow weakening of

kinetochore attachments in response to different kinds of error signals, potentially with different cell cycle timing. At least three other outer kinetochore components are regulated by both Aurora B and Mps1: the yeast Dam1 complex (Cheeseman et al., 2002; Shimogawa et al., 2006), its functional metazoan counterpart the Ska complex (Maciejowski et al., 2017; Redli et al., 2016), and the centromere protein E motor protein (Espeut et al., 2008; Kim et al., 2010). Phospho-proteomic analysis suggests that Ndc80 might be an Mps1 target in human cells (Maciejowski et al., 2017). Thus, outer kinetochore function could be regulated convergently by both kinases in multiple ways and across species.

Mps1-mediated phosphorylation of the Ndc80 tail causes a weakening of kinetochores similar to that caused by phosphomimetic substitutions at all seven Aurora B target sites within the tail (Sarangapani et al., 2013). This similarity is consistent with the simple view that Ndc80 tail phosphorylation provides rheostat-like control of kinetochore attachment strength (Zaytsev et al., 2015), with each phosphorylation event contributing equally and the resultant strength varying in proportion to the total number of unphosphorylated tail sites. However, a more complex view is suggested by the recent finding that some sites in the Ndc80 tail are dephosphorylated at metaphase while others remain phosphorylated throughout the cell cycle (DeLuca, 2017; Kucharski et al., 2021). Evidently not all phosphorylation sites are redundant, and therefore some might make differential contributions to chromosome segregation. In the future, it will be important to determine whether Aurora B and Mps1 act sequentially or simultaneously and whether any specific phosphorylation sites are more important for the regulation of kinetochore attachment strength to better understand why cells use multiple kinases to regulate the same sites (DeLuca et al., 2018; Kucharski et al., 2021; Zaytsev et al., 2015).

Although both Mps1 and Aurora B kinase can directly weaken kinetochores, mutating either one causes significant biorientation defects in vivo, which indicates that the two cannot fully compensate for one another to support this function. Such a requirement for both kinases in vivo, despite their similar direct effects in vitro, could arise because they each respond to different types of attachment errors or with different timing. Presumably it also reflects additional indirect effects that occur in cells when these kinases are mutated. If Aurora B is defective, for example, the opposing phosphatase PP1 prematurely localizes to kinetochores (Liu et al., 2010; Rosenberg et al., 2011), potentially causing dephosphorylation of Ndc80 and counteracting the role of Mps1 in error correction. Conversely, reducing Mps1 activity likely decreases Aurora B activity on kinetochores in budding yeast via regulation of the Bub1-shugoshin pathway (Storchová et al., 2011; Verzijlbergen et al., 2014; Yahya et al., 2020). Mps1 recruits the Bub1 kinase to kinetochores via phosphorylation of the Spc105 kinetochore protein (London et al., 2012; Peplowska et al., 2014; Shepperd et al., 2012; Yamagishi et al., 2012), which leads to Bub1-mediated phosphorylation of H2A and the subsequent recruitment of Sgo1 and Aurora B (Kawashima et al., 2010; Storchová et al., 2011; Verzijlbergen et al., 2014). Thus, reducing Mps1 activity could indirectly reduce Aurora B activity at kinetochores to a level that is insufficient for error correction. The Bub1-Sgo1 pathway also

contributes to kinetochore biorientation (Fernius and Hardwick, 2007; Peplowska et al., 2014), and it was recently reported that Mps1 phosphorylation of Spc105 is a key Mps1 target for kinetochore biorientation in vivo (Benzi et al., 2020). We found that Spc105 phosphorylation does not contribute to the Mps1-dependent weakening of reconstituted kinetochore–microtubule attachments in vitro. This result is consistent with our purified kinetochores lacking the Bub1-Sgo1 pathway and suggests that Mps1 plays multiple roles in achieving proper kinetochore–microtubule attachments in vivo. In the future, it will be important to understand the interplay between these pathways.

In summary, cells appear to rely on multiple kinetochore kinases as well as intrinsic kinase-independent mechanisms to avoid erroneous kinetochore–microtubule attachments during mitosis. While this complexity reflects sophisticated regulation, it also poses a major challenge for distinguishing direct contributions of specific kinases and substrates from indirect effects and, more generally, for developing a complete understanding of mitotic error correction. The combination of in vitro reconstitution with in vivo analysis that we used here should be useful in the future for further dissection of the vital processes by which dividing cells ensure the accuracy of chromosome segregation.

## Materials and methods

### Strain construction and yeast techniques
*Saccharomyces cerevisiae* strains used in this study are derivatives of SBY3 (W303) and described in Table S1. Standard media and microbial techniques were used, and yeast strains were constructed via standard genetic techniques (Rose et al., 1990). Construction of *stu2-3HA-IAA7* and *stu2^{\Delta cc}* (*pSTU2-stu2(\Delta 658-761::GDGAGL^{linker})-3V5*) are described in Miller et al. (2019, 2016), *DSN1-6His-3Flag* is described in Akiyoshi et al. (2010), *spc105-6A* is described in London et al. (2012), and *CEN8::lacO:TRP1* and *pCUP-GFP-LacI12* are described in Biggins et al. (1999), Miller et al. (2019), and Straight et al. (1996). Yeast strains with *ndc80-14A-3HA* were made by transformation with plasmid SB1848 (pSK1039, *pNDC80-ndc80-14A-3HA:KANMX*), a kind gift from Johannes Lechner (Biochemie-Zentrum der Universität Heidelberg, Heidelberg, Germany), or by crossing to derivative strains. Similarly, *ndc80-14D-3HA* yeast were made by transformation of plasmid SB1577 (pSK981, *pNDC80-ndc80-14D-3HA:KANMX*), a kind gift from Johannes Lechner, or by crossing to derivative strains. Strains containing *ndc80-11A-3HA* were made by transformation of plasmid SB3131, which was made by reverting T21, S37, and T74 in the *ndc80-14A* plasmid SB1848 (pSK1039; Lechner laboratory) via overlapping primer mutagenesis. *ndc80-8A-3HA* and *ndc80-8D-3HA* were made via standard cloning techniques from SB2412 (*Ndc80-3HA*). Briefly, pSB2412 and gBlocks (IDT Technologies) containing *ndc80-8A* (SB7002) or *ndc80-8D* (SB7001) were digested with BglII and BstEII and ligated using T4 DNA Ligase (NEB). Plasmids were fully sequenced. All plasmids are described in Table S2 and primers in Table S3. *Cdc20-AID* was a gift from Adele Marston's laboratory (Wellcome Center for Cell Biology, University of Edinburgh, Edinburgh, UK), and *dam1-3D(S257D,S265D,S292D)* was a

gift from the Barnes laboratory (Department of Molecular and Cell Biology, University of California, Berkeley, Berkeley, CA).

### Auxin-inducible degradation
To induce auxin-inducible degradation of the desired AID-tagged target protein, cells expressing a C-terminal fusion of an auxin-responsive protein (IAA7) in the presence of *TIR1*, which is required for auxin-induced degradation, were treated with 500 µM IAA (indole-3-acetic acid dissolved in DMSO; Sigma-Aldrich; Miller et al., 2016). For Fig. 6 G, auxin was added immediately after cells were released from α-factor.

### Serial dilution assay
The indicated yeast strains were grown overnight in yeast extract peptone dextrose (YPD) medium (2% glucose). The next day, the cells were diluted to $OD_{600}$ ∼1.0. A serial dilution (1:5) series was made in a 96-well plate, and cells were spotted onto YPD or YPD + 500 µM auxin (IAA; Sigma-Aldrich) plates. Plates were incubated for 1–3 d at 30°C unless otherwise indicated.

### Chromosome segregation and time course assays
Cells were grown at 23°C in YPD medium (2% glucose, 0.02% adenine). For the spindle assembly checkpoint assays, exponentially growing cells were arrested in G1 with 1 µg/ml α-factor for 3–4 h, washed 3 times, and then resuspended in medium lacking pheromone but containing 10 µg/ml nocodazole. Samples were collected at the indicated times. 1 µg/ml α factor was added to the cultures 40–50 min after G1 release to prevent cells from entering a second cell cycle.

To analyze chromosome segregation, exponentially growing *MATa* cells containing a LacI-GFP fusion and 256 *lacO* sequences integrated proximal to *CEN8* were arrested in G1 with 1 µg/ml α-factor. After 2.5 h, cells were washed and released into medium lacking pheromone but containing 500 µM IAA. α-Factor was readded ∼75 min after release to prevent entry into the subsequent cell cycle. 120 min after G1 release, aliquots of cells were fixed with 3.7% formaldehyde in 100 mM phosphate buffer (pH 6.4) for 10 min. Cells were then washed once with 100 mM phosphate buffer (pH 6.4), permeabilized, and stained with DAPI by resuspending in 1.2 M Sorbitol/1% Triton X-100/100 mM phosphate buffer (pH 7.5) containing 1 µg/ml DAPI (Molecular Probes) for 5 min. Cells were then resuspended in the same buffer lacking DAPI and applied to a coverslip treated with 0.5 mg/ml concanavalin A. Cells were imaged on a Deltavision Ultra deconvolution high-resolution microscope equipped with a 100×/1.4 PlanApo N oil-immersion objective (Olympus) with a 16-bit scientific complementary metal–oxide–semiconductor detector at 22°C. Cells were imaged in Z-stacks through the entire cell using 0.2-µM steps. All images were deconvolved using standard settings and analyzed with softWorX v7.2.1 (GE). Projections were made using a maximum-intensity algorithm. Composite images were assembled, and false coloring was applied with Adobe Photoshop.

To analyze phosphorylation of Ndc80 during the cell cycle, cells were initially grown at 23°C in YPD medium (2% glucose). Exponentially growing *MATa* cells were arrested in G1 with 1 µg/ml α-factor. After 2.5 h, cells were washed, released into medium

lacking pheromone, and shifted to 37°C to inactivate *mcd1-1*. 100-ml aliquots of cells were taken at 0, 60, and 90 min. Kinetochore purification assays were performed as described below for each time point, and phosphorylation of Ndc80 threonine-74 was analyzed via immunoblot.

For the nocodazole washout assay, yeast strains were grown overnight in YPD medium (2% glucose). The next day, the cells were diluted to $OD_{600}$ ~0.3 and cultured for 3 h. 500 µM IAA (Sigma-Aldrich) was added for 3 h to induce degradation of *cdc20-AID* and arrest cells in metaphase. During the final 30 min of the arrest, nocodazole was added to 10 µg/ml. Cells were washed twice in YPD and returned to culture. Time points were taken every 10 min for 1 h. Cells were fixed and DAPI stained as described above. Biorientation was scored in large-budded cells as either (a) two visible *CEN8::lacO* foci (bioriented) or (b) one visible *CEN8::lacO* focus (not bioriented). More than 200 cells were counted in three independent experiments, and the SEM is reported.

### Protein biochemistry

#### Kinetochore purification
Native kinetochore particles were purified from asynchronously grown *S. cerevisiae* cells grown in YPD medium (2% glucose), unless otherwise noted in the text, by modifying previous protocols (Akiyoshi et al., 2010; Miller et al., 2016). The *mps1-1* mutant kinetochores and native Dam1 complex were purified as previously described (de Regt et al., 2021 *Preprint*; Gutierrez et al., 2020). Protein lysates were prepared using a Freezer Mill (SPEX SamplePrep) submerged in liquid nitrogen. Lysed cells were resuspended in buffer H (25 mM Hepes, pH 8.0, 2 mM $MgCl_2$, 0.1 mM EDTA, 0.5 mM EGTA, 0.1% NP-40, 15% glycerol, and 150 mM KCl) containing phosphatase inhibitors (0.1 mM Na-orthovanadate, 0.2 µM microcystin, 2 mM β-glycerophosphate, 1 mM Na pyrophosphate, and 5 mM NaF) and protease inhibitors (20 µg/ml leupeptin, 20 µg/ml pepstatin A, 20 µg/ml chymostatin, and 200 µM PMSF). Lysates were ultracentrifuged at 98,500 *g* for 90 min at 4°C. Protein G Dynabeads (Invitrogen) were conjugated with an α-FLAG antibody (Sigma-Aldrich), and immunoprecipitation of Dsn1-6His-3Flag was performed at 4°C for 3 h. Beads were washed once with lysis buffer containing 2 mM DTT and protease inhibitors, three times with lysis buffer with protease inhibitors, and once in lysis buffer without inhibitors, and kinetochore particles were eluted by gentle agitation of beads in elution buffer (buffer H + 0.5 mg/ml 3FLAG Peptide [Sigma-Aldrich]) for 30 min at room temperature.

#### Mass spectrometry
Kinetochore purification assays were performed essentially as described above. After the final wash step, beads were rinsed twice in preelution rinse buffer (50 mM Tris, pH 8.3, 75 mM KCl, and 1 mM EGTA) and eluted in 50 µl elution buffer (0.2% Rapigest [Waters Corp.] and 50 mM ammonium bicarbonate) with gentle agitation for 30 min at 23°C. 10 µl of the sample was boiled in sample buffer and analyzed via silver stain (Fig. S4 A). The remaining 40 µl was snap frozen and prepared for mass spectrometry by reduction with DTT (10 mM in 100 mM ammonium bicarbonate) at 56°C for 45 min. The reduced protein

solutions were alkylated with 2-chloroacetamide (55 mM in 100 mM ammonium bicarbonate) and incubated in the dark at ambient temperature for 30 min. 250 ng Trypsin (Promega) was added to the solutions and incubated overnight at 37°C with mixing. Samples were acidified with 30% formic acid and mixed at room temperature for 1 h. Samples were spun down in a microfuge, and the supernatants were collected. All samples were desalted using ZipTip $C_{18}$ (Millipore) and eluted with 70% acetonitrile/0.1% trifluoroacetic acid. The desalted material was concentrated in a speed vacuum.

The generated peptide samples were analyzed with a Thermo Fisher Scientific Easy-nLC 1000 coupled in line with a Thermo Fisher Scientific Orbitrap Fusion mass spectrometer. Mass spectrometry data were analyzed using Thermo Fisher Scientific Proteome Discoverer v2.4, with Sequest HT as the protein database search algorithm. The data were searched against an SGD yeast (UP000000589 from Oct. 25, 2019) database that included common contaminants (cRAPome; Jan. 29, 2015). Searches were performed with settings for the proteolytic enzyme trypsin, and maximum missed cleavages was set to 2. The precursor ion tolerance was set to 10 ppm, and the fragment ion tolerance was set to 0.6 D. Variable modifications included oxidation on methionine (+15.995 D) and phosphorylation on serine, threonine, and tyrosine (+79.966 D). Static modifications included carbamidomethyl on cysteine (+57.021 D). Peptide validation was performed using Percolator, and peptide identifications were filtered to a 1% false discovery rate.

#### Immunoblot and silver stain analysis
Cell lysates were prepared as described in Kinetochore purification or by bead-beat pulverization (Biospec Products) with glass beads in SDS buffer. Standard procedures for immunoblot and SDS-PAGE were followed (Biggins et al., 1999). SDS-PAGE gels were transferred to 0.2 µm nitrocellulose membrane (Bio-Rad) using either the semi-dry method (Bio-Rad) or wet method (Hoefer). The anti-Mps1 antibodies were generated in rabbits against a recombinant Mps1 protein fragment (residues 440–764) of the protein by Genscript. The company provided affinity-purified antibodies that we validated by purifying kinetochores from yeast strains with Mps1 or Mps1-13Myc and confirming that the antibody recognized a protein of the correct molecular weight that migrated more slowly with the 13Myc epitope tags. We subsequently used the antibody at a dilution of 1:10,000. The phospho-specific T74 Ndc80 antibody was generated by Pacific Immunology against the phospho-peptide spanning resides 69–80 of Ndc80 (KRTRS-pT-VAGGTN-Cys) and subsequently affinity purified. Immunoblotting with this antibody was performed at a dilution of 1:1,000 with 1 µg/ml of nonphosphorylated competitor peptide (KRTRS-T-VAGGTN-Cys). The Ask1 antibody was affinity purified with recombinant GST-Ask1 from serum generated in rabbits against recombinant Dam1 complex and was used at a dilution of 1:5,000 (Gutierrez et al., 2020). The following commercial antibodies were used for immunoblotting: α-PGK1 (Invitrogen; 4592560; 1:10,000), α-FLAG (M2; Sigma-Aldrich; 1:3,000), α-Myc (7D10; Cell Signaling; 1:1,000), and α-HA (12CA5; Roche; 1:10,000). The α-Ndc80 antibody (OD4), used at 1:10,000, was a kind gift from

Arshad Desai (Department of Cellular & Molecular Medicine, University of California, San Diego, La Jolla, CA). Secondary antibodies used were donkey anti-rabbit antibody conjugated to HRP (GE Biosciences) at 1:10,000 or a sheep anti-mouse antibody conjugated to HRP (GE Biosciences) at 1:10,000. Antibodies were detected using SuperSignal West Dura Chemiluminescent Substrate (Thermo Fisher Scientific). Immunoblots were imaged with a ChemiDock MP system (Bio-Rad) or film. For silver stain analysis, protein samples were separated using precast 4–12% Bis-Tris Gels (Thermo Fisher Scientific) and stained using the Silver Quest Staining Kit (Invitrogen).

### Kinase assays

For radioactive kinase assays containing kinetochore particles (Fig. 2 A), kinetochores bound to beads were washed into kinase buffer (50 mM Tris-HCl, pH 7.5, 75 mM NaCl, 5% glycerol, 10 mM MgCl$_2$, 1 mM DTT, and 10 µM ATP) containing 66 nM γ-$^{32}$P-ATP with either 5 µM reversine (Sigma-Aldrich; R3904) or vehicle (DMSO) and incubated at 30°C for 30 min. Kinetochores were eluted by boiling in sample buffer containing SDS and analyzed via silver stain and Phosphorimager. In addition to the previously detected phosphorylation on Spc105, Ndc80, and Dsn1 (London et al., 2012), we detected phosphorylation on an additional unidentified protein (asterisk in Fig. 2 A), possibly due to altered SDS-PAGE conditions that allowed better resolution of kinetochore proteins. For kinase assays containing purified Ndc80 complex (Fig. 5 B), anti-HA immunoprecipitation was performed, and kinase assays were completed identically. Quantification of kinase assays was performed in Fiji (ImageJ) across three independent biological replicates. Intensity values from the Phosphorimager were normalized to Dsn1-His-Flag levels (Fig. 2 A) or Ndc80-3HA levels (Fig. 5 A), and the mean of the three replicates was used to calculate the change in phosphorylation between conditions.

### Phosphatase assay

Native kinetochore particles were purified (see Kinetochore purification) and kept bound to Dynabeads (Invitrogen). Beads were washed once with phosphatase buffer (buffer H, 1 mM MnCl), and then resuspended in phosphatase buffer with 200 U λ protein phosphatase (New England Biolabs) at 30°C for 20 min. Control samples contained phosphatase inhibitors. Kinetochores were eluted by boiling beads in sample buffer containing SDS, and phosphorylation was analyzed via immunoblotting.

### Biophysics

#### Laser trap instrument

Laser trap experiments were performed using a custom-built instrument based on a commercial inverted microscope (Nikon; TE2000U) that was modified for incorporation of a trapping laser (Spectra-Physics; J20I-BL-106C-NSI), a computer-controlled piezo specimen stage (Physik Instrumente; P-517.3CL), and a position-sensitive photodetector (Pacific Silicon Sensor; DL100-7-PCBA2). Details about the instrument have been described previously (Franck et al., 2010). To calibrate the instrument, position sensor response was mapped using the piezo stage to raster-scan a stuck bead through the beam, and trap stiffness was measured along the

two principal axes using the drag force, equipartition, and power spectrum methods. Force feedback was implemented with custom software written in LabView (National Instruments). During force measurements, bead-trap separation was sampled at 40 kHz, while stage position was updated at 50 Hz to maintain a force ramp rate of 0.25 pN • s$^{-1}$. Bead and stage position data were decimated to 200 Hz before storing to disk.

#### Bead functionalization and slide preparation for laser trap experiments

To link native kinetochore particles to laser trapping beads, we first functionalized streptavidin-coated polystyrene beads (0.56 µm in diameter; Spherotech) by incubating them with biotinylated anti-5His antibodies (Qiagen) and storing them for up to 3 mo with continuous rotation at 4°C in BRB80 (80 mM Pipes, 1 mM MgCl$_2$, and 1 mM EGTA, pH 6.9) supplemented with 8 mg • ml$^{-1}$ bovine serum albumin. Immediately before each experiment, the functionalized beads were decorated with kinetochore particles by incubating 6 pM anti-5His beads for 60 min at 4°C with purified kinetochore material, corresponding to Dsn1-His-Flag concentrations ranging between 2 to 4 nM. The resulting fraction of active beads capable of binding microtubules remained <50%, thus ensuring single-particle conditions (Akiyoshi et al., 2010; Sarangapani et al., 2013; Sarangapani et al., 2014).

Flow chambers (~10 µl volume) were made using glass slides, double-stick tape, and KOH-cleaned coverslips and then functionalized in the following manner. First, 15–20 µl of 10 mg • ml$^{-1}$ biotinylated bovine serum albumin (Vector Laboratories) was introduced and allowed to bind to the glass surface for 15 min at room temperature. The chamber was then washed with 100 µl of BRB80. Next, 25 µl of 1 mg • ml$^{-1}$ avidin DN (Vector Laboratories) was introduced, incubated for 3 min, and washed out with 100 µl of BRB80. GMPCPP-stabilized biotinylated microtubule seeds were introduced in BRB80 and allowed to bind to the functionalized glass surface for 3 min. The chamber was then washed with 100 µl of growth buffer (BRB80 containing 1 mM GTP and 1 mg • ml$^{-1}$ κ-casein). Finally, kinetochore particle-decorated beads (prepared as described in the previous paragraph) were diluted 8- to 10-fold into a solution of growth buffer containing 1.5 mg • ml$^{-1}$ purified bovine brain tubulin and an oxygen scavenging system (1 mM DTT, 500 µg • ml$^{-1}$ glucose oxidase, 60 µg • ml$^{-1}$ catalase, and 25 mM glucose) and then introduced into the flow chamber.

For the ATP (or mock) exposure experiments (Figs. 1, 2, 3, 5, S1, and S2), 6 mM ATP or ADP (conjugated with sodium salt; Sigma-Aldrich) was also added to the final dilution of kinetochore-decorated beads in growth buffer, just prior to introducing them into the flow chamber. In the case of phosphatase add-back experiments (Fig. 1), 1,200 units of λ-phosphatase were also added. For the reversine add-back experiments (Fig. 2), 5 µM reversine suspended in DMSO was also added. An equivalent volume of DMSO was used in the corresponding nonreversine control experiments. For some experiments using *mps1-1* kinetochores (Fig. S2 A), 2 nM of purified native Dam1 complex was added.

Immediately after introduction of the kinetochore-decorated beads, the edges of the flow chamber were sealed to prevent evaporation, and the time was set to 0 min. All laser trap

experiments were performed in temperature-controlled rooms, maintained at 23°C, for up to 90 min after the chamber was sealed.

### Rupture force measurements

For efficiency of data collection, beads that were already bound to microtubules (on the lattice, away from the dynamic tip) were usually chosen for measurements of rupture strength. Initially, the attachments were preloaded with a constant tensile force of 1–3 pN, which caused the lattice-bound beads to slide until reaching the microtubule plus end. Once they were at the end, we verified that the beads moved under the preload force at a rate consistent with that of microtubule growth or shortening. The laser trap was subsequently programmed to ramp the force at a constant rate (0.25 pN • s⁻¹) until the linkage ruptured or the load limit of the trap was reached (~23 pN under the conditions used here). Fewer than 5% of all trials ended in detachment during the preload period before force ramping began, while 0–15% reached the load limit (depending on the conditions tested). These out-of-range events were included in the median force calculations and the survival probability distributions. Under selected conditions, we also tested beads that were floating freely in solution to estimate the fraction of active beads that were capable of binding microtubules. When beads decorated with WT kinetochore particles (from SBY8253) were tested in the absence of adenosine, 6 of 30 beads tested (20%) bound to microtubules. Similarly, when identically prepared beads were tested in the presence of ADP, 5 of 22 (23%) bound microtubules. However, when the beads were tested in the presence of ATP, only 3 of 27 tested (11%) bound to microtubules, suggesting that the binding activity of the kinetochore particles (like their rupture strength) is reduced upon exposure to ATP. We found no statistically significant difference in the rupture strengths of prebound versus free beads. Most of the measurements (>98%) were obtained using prebound bead–microtubule pairs. All rupture force data are presented as bar graphs, which display median values with ±95% confidence intervals estimated by bootstrapping, and also as survival plots, which display complete distributions of the measured rupture forces without binning or smoothing (Figs. 1, 2, 3, 5, S1, and S2). All the individual measured rupture force values, the numbers of trials that reached the load limit of the trap without rupture, and the survival probability curves for every measurement condition, are provided in Table S5. Statistical comparisons of rupture strengths between different measurement conditions were made using log-rank tests (Table S4), which are nonparametric tests (and thus do not assume normally distributed data) that compare pairs of survival distributions directly to one another. P values computed from these log-rank tests for all the possible pairwise comparisons within each figure are provided in Table S5.

### Online supplemental material

Fig. S1 shows control experiments related to Fig. 1. Fig. S2 shows control experiments related to Figs. 2 and 3. Fig. S3 shows control experiments related to Fig. 4. Fig. S4 shows control experiments related to Fig. 5. Information regarding yeast strains, plasmids, and oligonucleotides used in this study are provided in Table S1, Table S2, and Table S3. Table S4 summarizes the median rupture force values for each kinetochore type or condition. All the individual rupture force values, Kaplan–Meier survival probability estimates (with 95% confidence intervals), numbers of trials that reached the load limit of the trap without rupture, and other statistics (e.g., P values from log-rank tests) are included in Table S5. Mass spectrometry raw data are provided in Table S6.

## Acknowledgments

We are grateful for feedback and critical reading of the manuscript from members of the Biggins and Asbury laboratories, as well as the Seattle Mitosis Group. We are grateful to Arshad Desai, Adele Marston, Georjana Barnes, the Lechner Lab, and Abraham Gutierrez for contributing reagents and to Lisa Jones for her help in obtaining mass spectrometry data.

L.B. Koch was supported by a National Science Foundation Graduate Research Fellowship (DGE-1256082). This work was supported by a Packard Fellowship 2006-30521 (to C.L. Asbury), National Institutes of Health grants R01GM079373, P01GM105537, and R35GM134842 (to C.L. Asbury) and R01GM064386 (to S. Biggins), and by the Genomics and Scientific Imaging and the Proteomics and Metabolomics Shared Resources of the Fred Hutchinson/University of Washington Cancer Consortium (P30 CA015704). S. Biggins is an investigator of the Howard Hughes Medical Institute.

The authors declare no competing financial interests.

Author contributions: Conceptualization (K.K. Sarangapani, L.B. Koch, C.R. Nelson, C.L. Asbury, S. Biggins), data curation (K.K. Sarangapani, L.B. Koch, C.R. Nelson), formal analysis (K.K. Sarangapani, L.B. Koch, C.R. Nelson, C.L. Asbury, S. Biggins), funding acquisition (L.B. Koch, C.L. Asbury, S. Biggins), investigation (K.K. Sarangapani, L.B. Koch, C.R. Nelson), project administration (C.L. Asbury, S. Biggins), resources (K.K. Sarangapani, L.B. Koch, C.R. Nelson, C.L. Asbury, S. Biggins), software (C.L. Asbury), supervision (C.L. Asbury, S. Biggins), validation (K.K. Sarangapani, L.B. Koch, C.R. Nelson, C.L. Asbury, S. Biggins), visualization (K.K. Sarangapani, Lori Koch, C.R. Nelson, C.L. Asbury, S. Biggins), writing–original draft (L.B. Koch and S. Biggins), writing–review and editing (K.K. Sarangapani, L.B. Koch, C.R. Nelson, C.L. Asbury, S. Biggins).

Submitted: 24 June 2021

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

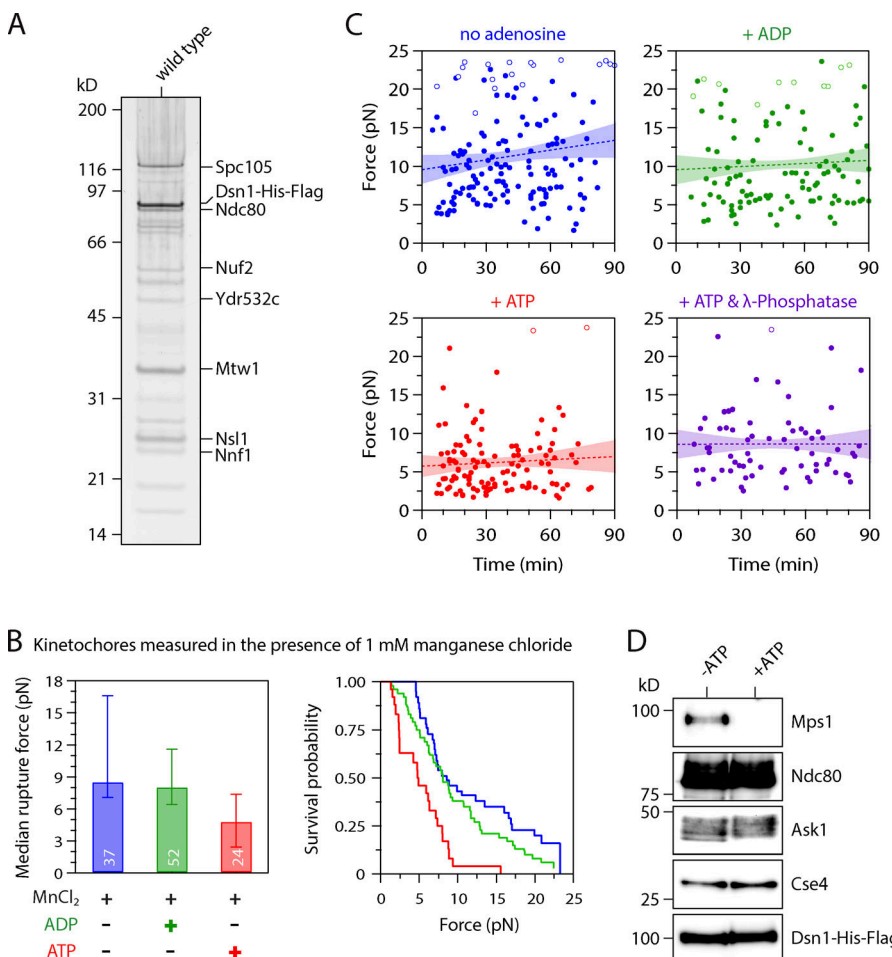

Figure S1. **Control experiments related to Fig. 1.** **(A)** Kinetochore material purified by immunoprecipitation of Dsn1-His-Flag from WT cells (SBY8253) visualized by silver stain after SDS-PAGE. **(B)** Median rupture strengths (left) and corresponding survival probability distributions (right) for WT kinetochores (from SBY8253) in the presence of 1 mM manganese chloride (MnCl₂), which was included in the experiments with λ-phosphatase (Fig. 1 D) because it is necessary for phosphatase activity. The kinetochores under these conditions were weakened upon exposure to ATP by an amount similar to that in the absence of manganese chloride. Values inside bars indicate numbers of events for each condition. Error bars represent ± 95% confidence intervals calculated by bootstrapping. P values for all pairwise strength comparisons (from log-rank tests) are provided in Table S5. **(C)** The ATP-dependent weakening reaction is completed within minutes. Individual rupture force values (solid circles) are plotted against the time elapsed since the kinetochore-decorated beads were prepared as indicated (without adenosine, or mixed with ATP, ADP, or ATP + λ-phosphatase). Right-censored events that reached the load limit of the laser trap before rupture are also plotted (open circles). The slopes of lines fitted to all the data (dotted lines, with 95% confidence intervals shown) were not significantly different from zero (0.42 ± 0.45, 0.014 ± 0.045, 0.014 ± 0.040, and 0.000 ± 0.049 pN • min⁻¹, respectively), indicating that ATP-dependent weakening occurred during the ~10-min slide preparation, with no significant weakening thereafter. **(D)** Kinetochores purified by immunoprecipitation of Dsn1-His-Flag from WT cells (SBY8253) were equilibrated in kinase buffer with or without ATP (200 µM), incubated at 30°C for 20 min, and then eluted and analyzed by immunoblotting for the indicated kinetochore components.

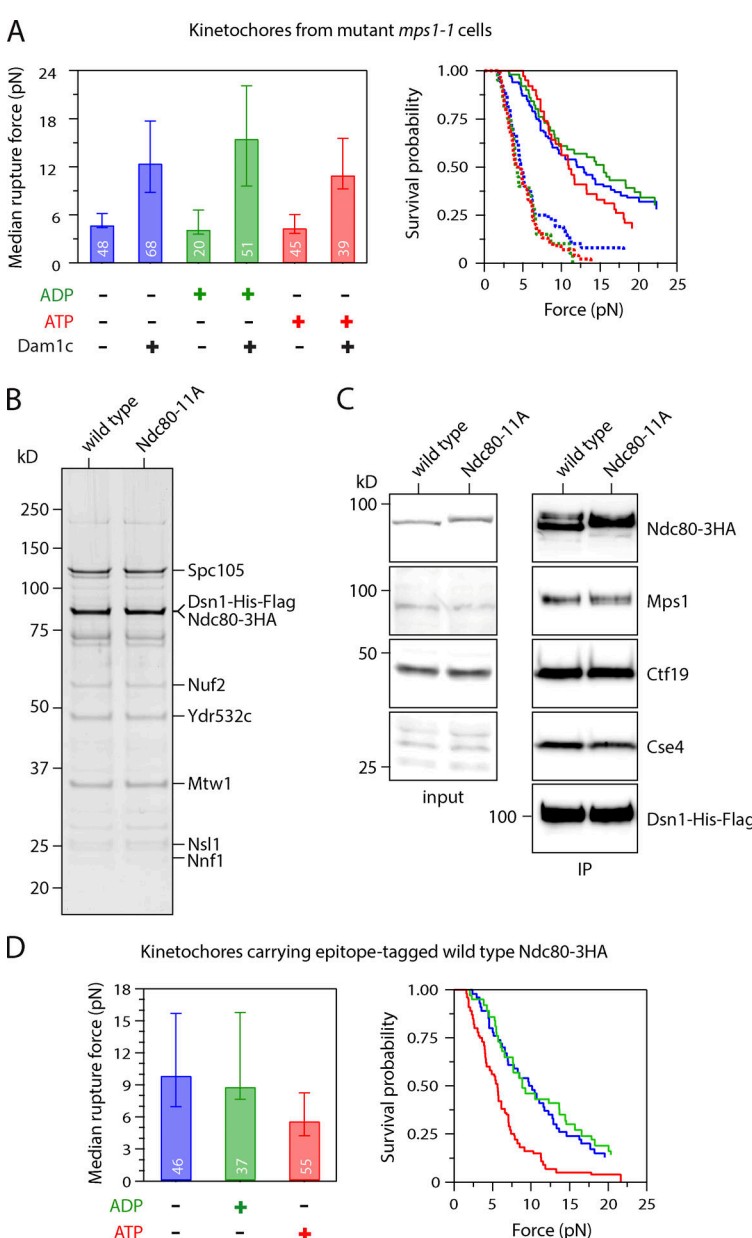

Figure S2. **Control experiments related to Figs. 2 and 3. (A)** Median rupture strengths (left) and corresponding survival probability distributions (right) for mutant *mps1-1* kinetochores (from SBY8726) measured under indicated conditions. The *mps1-1* kinetochores are intrinsically weaker than WT kinetochores due to a lack of copurifying Dam1 complex (Dam1c), which can be added exogenously to increase their strength. Irrespective of whether exogenous Dam1 complex was added, the mutant *mps1-1* kinetochores were not significantly weakened by exposure to ATP. **(B)** Kinetochore material purified by immunoprecipitation of Dsn1-His-Flag from cells carrying epitope-tagged WT Ndc80-3HA (SBY11808) and from mutant cells carrying phospho-deficient Ndc80-11A-3HA (SBY19380) visualized by silver stain after SDS-PAGE. **(C)** Similar levels of Mps1 copurify with WT and Ndc80-11A kinetochores. Kinetochore material was purified by immunoprecipitation (IP) of Dsn1-HIs-Flag (from SBY11808 and SBY19380) and analyzed by immunoblotting. No Dsn1-His-Flag blot is shown for the input material because Dsn1 was undetectable in whole-cell lysates, due to an overlapping cross-reactive band. **(D)** Control experiments related to Fig. 3 F. Median rupture strengths (left) and corresponding survival probability distributions (right) for kinetochores carrying epitope-tagged WT Ndc80-3HA (from SBY11808). These kinetochores were weakened upon exposure to ATP by an amount similar to kinetochores carrying untagged WT Ndc80. Values inside bars in A and D indicate numbers of events for each condition. Error bars in A and D represent ± 95% confidence intervals calculated by bootstrapping. P values for all pairwise strength comparisons (from log-rank tests) are provided in Table S5.

Figure S3.   **Control experiments related to Fig. 4.** The phospho-mimetic ndc80-14D allele is lethal, and its lethality is not rescued by deletion of the checkpoint gene *MAD2*. However, suppressors do arise after many days of growth. The indicated strains (SBY15149, SBY9453, SBY15139, SBY17991, SBY17993, SBY15087, and SBY11334) were tested using a plasmid shuffle assay, where growth on 5-FOA plates indicates viability in the absence of a covering copy of WT *NDC80*. Fivefold serial dilutions were plated onto YPD, −uracil, or 5-FOA plates and grown at 23°C for 2 d. After 7 d of growth on 5-FOA, a small number of colonies grow from the *ndc80-14D* (SBY9453), *ndc80-14D bub1Δ* (SBY15139), and *ndc80-14D mad3Δ* (SBY11334) strains.

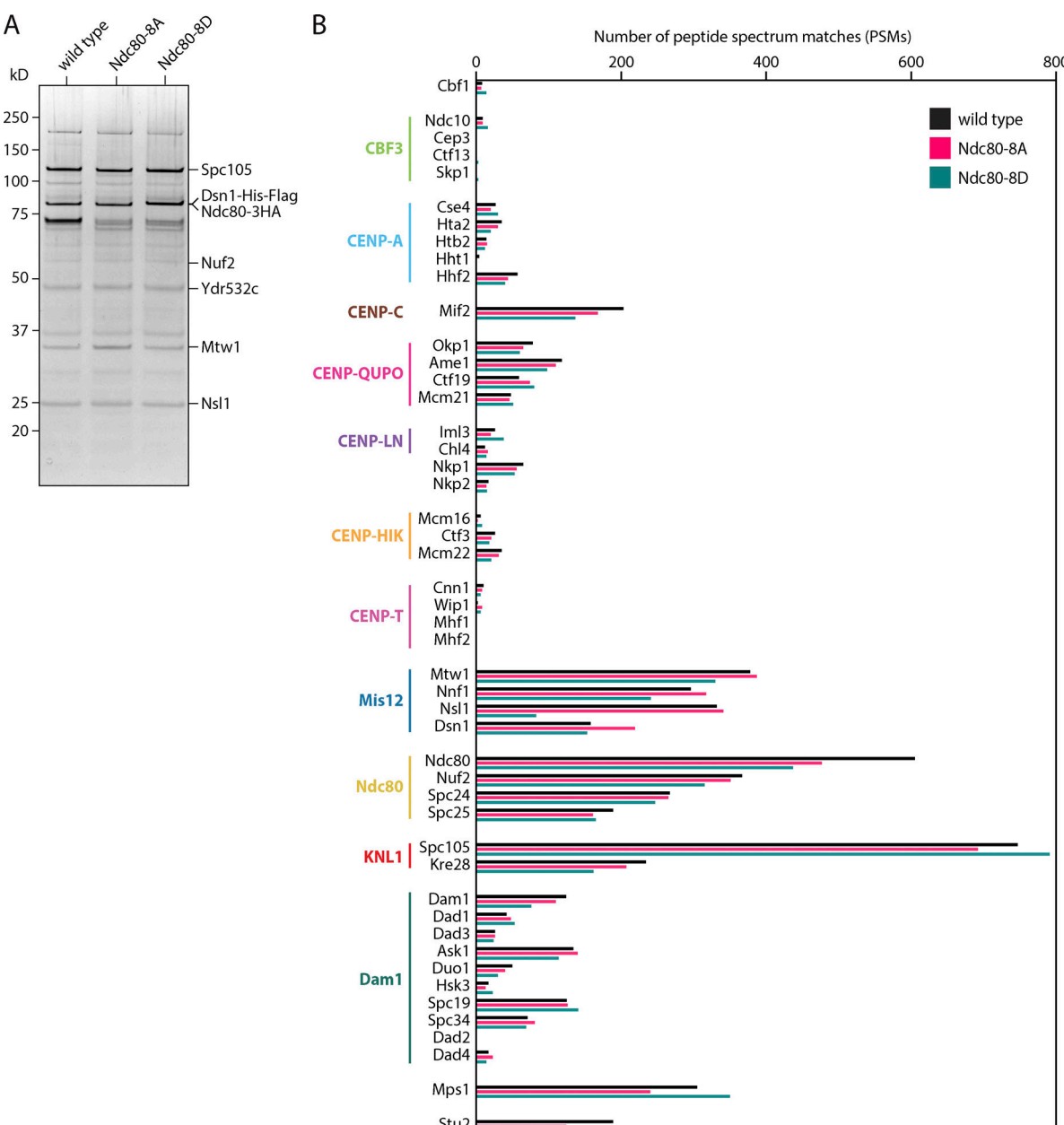

Figure S4. **Control experiments related to Fig. 5. (A)** Kinetochore material purified by immunoprecipitation of Dsn1-His-Flag from cells carrying epitope-tagged WT Ndc80-3HA (SBY19186), phospho-deficient Ndc80-8A-3HA (SBY19855), or phospho-mimetic Ndc80-8D-3HA (SBY19877), visualized by silver stain after SDS-PAGE before mass spectrometry analysis. **(B)** Results from mass spectrometry analysis of the same material visualized in A. Similar numbers of peptide spectrum matches were detected for the indicated kinetochore proteins across each of the three samples.

**Provided online are six tables. Table S1 lists yeast strains used in this study. Table S2 lists plasmids used in this study. Table S3 lists primers used in this study. Table S4 summarizes laser trap results. Table S5 shows rupture force values, Kaplan–Meier estimates, numbers of trials that reached load limit without rupture, and other statistics. Table S6 shows mass spectrometry raw data.**

