## [Peer Review File · The Journal of Cell Biology]

Kinetochores-bound Mps1 regulates kinetochores-microtubule attachments via Ndc80 phosphorylation

Krishna Sarangapani, Lori Koch, Christian Nelson, Charles Asbury, and Sue Biggins

Corresponding Author(s): Sue Biggins, Fred Hutchinson Cancer Research Center and Charles Asbury, University of Washington School of Medicine

Review Timeline:

Submission Date:	2021-06-24
Editorial Decision:	2021-07-26
Revision Received:	2021-08-03
Editorial Decision:	2021-08-31
Revision Received:	2021-09-07

Monitoring Editor: Aaron Straight

Scientific Editor: Dan Simon

Transaction Report:

DOI: <https://doi.org/10.1083/jcb.202106130>

Revision 0

Review #1

1. How much time do you estimate the authors will need to complete the suggested revisions:

Estimated time to Complete Revisions (Required)

(Decision Recommendation)

Between 1 and 3 months

2. Evidence, reproducibility and clarity:

Evidence, reproducibility and clarity (Required)

The authors use a combination of Dsn1-Flag kinetochore purification from yeast extracts and laser trapping experiments (as in a number of previous studies), to study the effect of Mps1-dependent phosphorylation on reconstituted kinetochore-microtubule attachments in vitro. They complement this analysis with genetic experiments characterizing the effects of non-

Mps1 phosphorylatable mutants on checkpoint activity and chromosome segregation in yeast.

The authors had previously shown that Mps1 is the major kinase activity that copurifies with Dsn1-Flag in their purification scheme. They now investigate the effect of adding ATP and thereby allowing Mps1 phosphorylation in the reconstituted system. They show that addition of ATP decreases the rupture force of kinetochore-microtubule attachments, meaning it weakens the strength of the attachment. This effect can be negated either by inhibiting Mps1 with reversine, or by providing kinetochores in which the Mps1 phosphorylation sites on Ndc80 (most of them in the N-terminal tail) have been mutated to alanine. Thus, similar to the activity of Ipl1, Mps1 phosphorylation of the Ndc80 N-tail (which is known to be important for full MT affinity) weakens kinetochore-microtubule attachments.

Cellular experiments demonstrate that non-Mps1 phosphorylatable Ndc80 14-A mutants have a functional mitotic checkpoint (contrary to previous claims by Kemmler et al., 2009), but show synthetic sickness with *stu2* alleles that are involved in error correction.

****Major points:****

Within the framework of this experimental setting, the study as presented is logical and clear. The conclusions regarding the effect of Mps1 in this reconstituted system are overall well supported by the data. I have a couple of major and some minor points that can further improve data interpretation and should therefore be considered:

1. In previous publications (e.g. Gutierrez et al., Current Biology 2020), the authors have reported that the Dam1 complex, an established Mps1 substrate, is required for full attachment strength in this system. Are the effects of Mps1-dependent Ndc80 phosphorylation and Dam1 independent from one another? For example would *dad1-1* or non Cdk1 phosphorylatable Dam1 complex further reduce the rupture force in ATP? Or does Mps1 phosphorylation affect, for example, Dam1 binding to Ndc80?

2. What is the effect of ATP on initial binding events? Are there differences in the fraction of beads that spontaneously attach laterally at the start of the experiment?

This may allow to draw conclusions whether any kind of binding or specifically force-generating end-on attachments are affected by ATP.

3. Ndc80-8D has low attachment strength, consistent with lowered MT affinity of the phospho-mimetic Ndc80 tail. Interestingly, Supplementary Figure S4B shows that the amount of Cse4 in the pull-down western appears substantially reduced in 8D vs 8A or wt. Is the amount of co-purified inner

kinetochore affected in this mutant? This may be an alternative explanation for decreased attachment strength, for example if the fraction of "full" or "complete" kinetochores may be reduced. Could this also happen upon inclusion of ATP?

****Minor points:****

page 13 (heading): "Weakening occurs via phosphorylation...". Probably good to mention what is weakened ("Weakening of kinetochore-microtubule attachments occurs via phosphorylation...").

page 14/Figure5C: Median Rupture Force for Ndc80-8D is 4.8 pN according to the text. In the graph it looks like >5 pN.

page 23: comma missing between T21 S37 and T47 (should be T21, S37 and T47)

page 24/25: different spelling of G1 (sometimes with subscript)

page 24/25: ug instead of μg

page 28: Figure 5B instead of Figure 5A

Figure 6A: Lambda-Phosphatase treatment for 20 minutes according to figure legend and 30 minutes according to Material and Methods section.

Figure 6E: One should not draw any conclusions from the anti-phospho T47 blot here, the quality is simply too poor to allow a statement regarding an mps1-1 effect

Figure 6: Labelling T47P misleading (Proline substitution?, use pT47

instead)

Figure 6F: Make clear in the labelling that a *stu2*-AID background is used here, makes it easier to understand why Auxin is used here.

how specific is reversine for yeast Mps1? I have not seen any data on this in previous publications.

additional genetic interactions might be informative, if Ndc80-8D has weakened attachments, it may have synthetic effects with other mutants (*dam1*?), conversely, *ndc80-8A* may show genetic interactions with *ipl1* alleles, for example.

3. Significance:

Significance (Required)

The study adds to the characterization of the effects of Mps1 kinase on kinetochore-microtubule attachments and characterizes the cellular phenotypes of non-Mps1 phosphorylatable Ndc80 mutants. The major conceptual point that Mps1 phosphorylation can weaken kinetochore-microtubule interactions and thereby contributes to error correction in a manner similar to Ipl1 has previously been made in the literature. Maure et al., (Tanaka lab, 2007, Current Biology) have characterized the effects of *mps1* mutant alleles on biorientation of authentic chromosomes and on replicated/unreplicated mini-chromosomes. In particular the experiments with unreplicated mini-chromosomes have revealed less frequent detachment in *mps1* mutants, demonstrating that Mps1 activity is required to release attachments that are not under tension.

Another benefit of this study is that it puts the Kemmler 2009 EMBO J. paper into perspective and corrects some of its claims. In particular the notion of sustained checkpoint activation in the Mps1 phospho-mimetic Ndc80-14D mutant, whose lethality was claimed to be rescued by checkpoint deletion. It is confirmed here that the allele is lethal, but cannot be alleviated by simultaneous checkpoint deletion. Conversely, the Ndc80-14A mutant is shown to have a functional checkpoint. One could argue that since

the publication of the Kemmler paper, the idea of requirement of Mps1 phosphorylation on Ndc80 for checkpoint activity has not gained any traction in the field, but it's still useful for the field to put some of these earlier claims into perspective. The paper will therefore be interesting to researchers working on mechanisms of chromosome segregation and error correction.

From my background I cannot comment on technical details of the biophysical force spectroscopy experiments (laser trapping), but I have no reason to doubt that the authors accurately report their findings.

Review #2

1. How much time do you estimate the authors will need to complete the suggested revisions:

Estimated time to Complete Revisions (Required)

(Decision Recommendation)

Between 1 and 3 months

2. Evidence, reproducibility and clarity:

Evidence, reproducibility and clarity (Required)

This paper focusses on the mechanisms underlying chromosome biorientation in mitosis, an essential process that warrants equal chromosome segregation to the dividing cells. Correction of improper kinetochore-microtubule attachments relies on two conserved protein kinases, Aurora B and Mps1, that detach kinetochores that are not under tension in order to provide them with a second opportunity to establish bipolar connections. In vivo, Aurora B and Mps1 have intertwined functions and share some common targets. For this reason, despite the large body of literature on the subject, their precise roles in chromosome biorientation have been difficult to tease apart.

The authors take advantage of an in vitro reconstitution assay that they previously published (Akyioshi et al., 2010) to identify the critical target(s) of Mps1 in weakening kinetochore-microtubule connections. The assay uses kinetochore particles purified from budding yeast cells that bear Mps1 but are notably deprived of Aurora B. Upon addition of ATP to activate the co-purified kinases (e.g. Mps1), kinetochores are added to coverslip-anchored microtubules to which they attach laterally. Through a laser trap, kinetochores are brought to the microtubule plus-end and pulled with increasing force until the kinetochore detaches, which allows measurements of the average rupture forces that reflect the strength of the attachments. The approach is straightforward and potentially very powerful, first because it provides a simplified experimental set-up in comparison to the cellular context, and second because it directly measures the impact of protein phosphorylation on the strength of attachments.

The authors convincingly show that Mps1-dependent phosphorylation of the N-terminal part of Ndc80 significantly weakens the strength of kinetochore-microtubule attachments in vitro, while phosphorylation of other known Mps1 targets, such as Spc105, does not seem to have an effect. Eight phosphorylation sites in Ndc80, which were previously identified as Mps1-dependent phosphorylation sites (Kemmler et al., 2009), are shown to be critical to destabilise kinetochore-microtubule attachments in the in vitro reconstitution assays. The authors also present evidence for a moderate involvement of Ndc80 phosphorylation by Mps1 in correcting improper attachments in vivo, suggesting that additional mechanisms are physiologically relevant for error correction.

The experiments are mostly well designed, the data are solid and support the main conclusions. However, to my opinion additional experiments could be performed, as outlined below, to strengthen the physiological relevance

of the main findings and corroborate some of the conclusions.

****Major points:****

1. Given the partially overlapping function of Mps1 and Ipl1 (Aurora B) in error correction, the ndc80-8A mutant should display synthetic growth and chromosome mis-segregation defects with ipl1 temperature-sensitive alleles. Conversely, the ndc80-8D mutant should suppress the lethality at high temperatures of mps1-3 mutant cells, which were recently shown to be defective in chromosome biorientation (Benzi et al., 2020). Finally, chromosome mono-orientation could become apparent in ndc80-8A cells upon a transient treatment with microtubule-depolymerising drugs, which should amplify the cellular need for error correction.

2. The authors show that Mps1-dependent phosphorylation of Ndc80 is not involved in the spindle assembly checkpoint, a conclusion that contradicts a previous report (Kemmler et al., 2009). They also find, in contrast with the same report, that the lethal phenotype of the ndc80-14D phospho-mimetic mutant cannot be rescued by disabling the spindle checkpoint. In my opinion, Kemmler et al. convincingly showed, through a number of different experimental approaches, that ndc80-14D cells die because of spindle checkpoint hyperactivation. Not only deletion of checkpoint genes was shown to rescue the lethality, but re-introduction of a wild type copy of the deleted checkpoint gene reinstated lethality. Thus, the explanation invoked here that spontaneous suppressing mutations could underlie the viability of ndc80-14D SAC-deficient mutants is not consistent with the published observations. A thorough examination by the authors of the phenotype of ndc80-14D cells in their hands should be carried out to support these conflicting conclusions. If authors find that ndc80-14D cells actually die because of chromosome mono-orientation, then this would highlight an important function for some or all the six additional phosphorylation sites, relative to the ndc80-8D mutant, for chromosome biorientation in vivo.

3. The conclusion that Spc105 phosphorylation by Mps1 is not required for the Mps1-mediated weakening of kinetochore attachments in vitro is based on the comparison between kinetochore particles bearing wild type, untagged Spc105 and particles bearing non-phosphorylatable Spc105-6A tagged at the C-terminus with twelve myc epitopes. Thus, the presence of the tag could obliterate the effects of the mutations in the phosphorylation sites by destabilising kinetochore-microtubule attachments in the presence of ATP. Consistent with this conclusion, Spc105-6A-12myc-bearing kinetochores withstand lower rupture forces than Spc105-bearing kinetochores upon ATP addition. Furthermore, Spc105-6A-12myc kinetochore particles show an interacting protein at MW above 150 KD that is not present in wild type particles (Fig. S2A), suggesting that either the tag or the mutations might affect kinetochore composition. Thus, this set of experiments should be repeated using Spc105-6A kinetochore particles lacking the tag.

4. In general, it would have been informative to complement the data presented here with a mass spec analysis of the composition of kinetochore particles, at least for the experiments that are most relevant to the

conclusions. For instance, the composition of the Ndc80-8A kinetochore particles is assumed to be similar to that of wild type kinetochores based on gel silver staining (Fig. S4A; note also that ndc80-8A particles are compared to ndc80-8D particles and not to wild type particles). However, the authors previously showed that kinetochore particles purified from dad1-1 mutant cells (affecting the Dam1 complex) have an apparently identical composition to particles purified from wild type cells by silver staining, yet they display significantly lower resistance to the rupture strength in vitro (Akyiوشي et al., 2010). What is the status of the Dam1 complex (or other kinetochore subunits) in kinetochores purified from ndc80-8A/-8D or spc105-6A cells relative to wild type kinetochore particles?

****Minor comment:****

I believe that the right reference for the sentence in the Discussion "If Aurora B is defective, for example, the opposing phosphatase PP1 prematurely localizes to kinetochores" is Liu et al. 2010.

3. Significance:

Significance (Required)

Although the experiments are well designed and the conclusions are mainly supported by the data, the question arises as to what extent the in vitro assays recapitulate, at least partly, what happens in vivo. An emblematic example is the involvement of Spc105 in the error correction pathway. The Biggins lab previously showed that Spc105 phosphorylation by Mps1 and subsequent Bub1 recruitment is not only essential for the spindle assembly checkpoint, but is also crucial for chromosome segregation in vivo, as shown by slow-growth phenotype and aneuploidy of the spc105-6A non-phosphorylatable mutant (London et al., 2012). Additionally, a recent paper showed that Spc105 is a crucial Mps1 target in chromosome biorientation (Benzi et al., 2020).

In sharp contrast, the ndc80-8A mutant, which in vitro completely erases the ability of Mps1 to destabilise kinetochore-microtubule attachments, displays no growth defects in otherwise wild type cells and only modestly enhances chromosome mis-segregation in a mutant affecting an intrinsic correction pathway (stu2ccΔ). The N-terminal part of Ndc80 (aa 1-116) containing the aforementioned eight phosphorylation sites can even be deleted altogether without any consequence on cell viability (Kemmler et al., 2009). Thus, although the in vitro assays presented here produced clear-cut and reproducible results, their physiological relevance in vivo remains unclear.

Left apart this criticism, the manuscript has several merits outlined above and will be of interest for people working in the fields of chromosome segregation, kinetochore assembly, spindle assembly checkpoint, etc.

Expertise of this reviewer: mitosis and related checkpoints

Review #3

1. How much time do you estimate the authors will need to complete the suggested revisions:

Estimated time to Complete Revisions (Required)

(Decision Recommendation)

Between 1 and 3 months

2. Evidence, reproducibility and clarity:

Evidence, reproducibility and clarity (Required)

Sarangapani, Koch, Nelson et al. applied a combination of in vitro biophysical assays with purified kinetochore particles and in vivo analyses to investigate the contribution of Mps1 kinase to kinetochore-microtubule (KT-MT) attachment stability and error correction.

The manuscript is well written and the authors nicely highlight the facts that 1) the focus of the field has long been on the contribution of Aurora kinases (Ipl1 in budding yeast) to attachment stability and error correction, and 2) it has been difficult to assess the relative contributions of Aurora versus Mps1 kinases in cell-based experiments. The authors note that their KT particle assay is uniquely positioned to address this gap in our understanding and to specifically isolate the contribution of Mps1 to attachment stability in vitro. The findings are well-presented and quite convincing although I have several comments that should be addressed to strengthen the central conclusion that this work has isolated the contribution of Mps1 in their assays.

****Major points:****

1) I think it is important to note that reversine is not specific for Mps1 kinase - although it is typically presented as such in the field. It was initially identified as an Aurora kinase inhibitor (IC₅₀: ~25nM (Aurora B) - 900nM (Aurora A)) that turned out to be an even more potent Mps1 inhibitor (IC₅₀ ~6nM). I have concerns that the in vitro assays were done with 5 uM reversine - a concentration so high that it could certainly inhibit any Ipl1 that is present (see comment 3 below) and possibly even inhibit Bub1 activity as Santaguida et al. (JCB, 2010) measured an IC₅₀ >1uM for Bub1 inhibition. It is important to complement/confirm the chemical inhibitor experiment by repeating the rupture assays +/- ATP in KT particles purified from the mps1-1 strain (shown in Figure 6).

2) If the ATP-mediated reduction on rupture force is lost in the mps1-1 KT

particles, which will also lack Bub1 kinase, then preserving the ATP-dependent reduction in rupture force from KT particles purified from the Bub1delta mutant strain would be strong evidence that the contribution of Mps1 kinase has been disentangled from other kinases in this assay.

3) Recent work has shown that Sli15-Ipl1 interacts with and is recruited to KTs by the COMA complex (Rodriguez et al., *Curr Biol*, 2019 and Fischbock-Halwachs et al., *eLife* 2019) and that this population of Ipl1 is important for accurate chromosome segregation as also shown 10 years prior by Knockleby and Vogel (*Cell Cycle*, 2009). I realize that this group previously showed (London et al., *Curr Biol*, 2012) that phosphorylation of KT particles was not affected when purified from the *ipl1-321* mutants, but in light of the recent findings how sure are the authors that there is not any Sli15-Ipl1 in the preparations? I think commenting on this would be worthwhile.

4) Since the interplay between Mps1 and Aurora B are central to this story, the authors should expand upon the sentence on page 5 reading "While there is some evidence that Mps1 regulates Aurora B activity (Jelluma et al., 2010; Saurin et al., 2011; Tighe et al., 2008), significant data suggests it has an independent role in error correction and acts downstream of Aurora B (Hewitt et al., 2010; Maciejowski et al., 2010; Maure et

al., 2007; Meyer et al., 2013; Santaguida et al., 2010)." I am not entirely convinced that the *in vivo* experiments presented here differentiate as to whether Mps1 is upstream from Ipl1 or whether they are acting independently? For example, phosphorylation of T74 looks to be completely lost in figure 6E (although it's difficult to tell since the blot for T74P is very smeary). If they are acting independently in error correction then Ipl1 should still be able to phosphorylate T74 in this condition. However, if the P-T74 really is lost completely in the *mcd1-1* cells then this suggests to me that Ipl1 is downstream of Mps1 in this live cell error correction assay.

****Other points:****

1) On p.8 "a median strength of 7.5 pN, similar to untreated and ADP-treated kinetochores". Similar is vague so I'm curious as to whether there a statistically significant difference between this and the 9.8 pN and 8.7 pN measured in the other conditions. If so this could be explained by partial dephosphorylation with the phosphatase.

2) On p.19 the authors note that Aurora A phosphorylates Ndc80 tail during mitosis. Ye et al. (*Curr Biol*, 2015) also showed that Aurora A can phosphorylate Aurora B sites and that this activity "converges" at the tail to weaken attachments during error correction.

3) Optional: I am curious as to whether the addition of ATP to the Ndc80-8D particles further reduces the rupture force. If so then other sites may also be in play.

4) Please comment on why MnCl₂ is used in the rupture assays in Figure S1. I saw no mention of this in the main text.

5) Consider moving S2 A and B to Figure 3 C and D. This is an interesting result and would go well in the main figure next to the significantly reduced rupture force measurements for the 6A mutant so the reader doesn't have to dig into the supplemental for the data providing this reasonable explanation for the rupture force result.

3. Significance:

Significance (Required)

The significance of this relates to focusing on an important phenomenon - error correction - and in looking beyond the traditional focus of the field on Aurora kinases to Mps1 kinase, which is largely implicated in checkpoint signaling. Disentangling the contributions of these two players is an important advance.

The work will be of interest to audiences interested in: kinases, cell division, checkpoints, kinetochore biology, biophysics

The above areas of interest overlap with my expertise.

We are grateful to the reviewers for their thoughtful comments and propose the following experiments or clarifications listed below (blue) in a revised manuscript.

Reviewer #1 (Evidence, reproducibility and clarity (Required)):

The authors use a combination of Dsn1-Flag kinetochore purification from yeast extracts and laser trapping experiments (as in a number of previous studies), to study the effect of Mps1-dependent phosphorylation on reconstituted kinetochore-microtubule attachments *in vitro*. They complement this analysis with genetic experiments characterizing the effects of non-Mps1 phosphorylatable mutants on checkpoint activity and chromosome segregation in yeast.

The authors had previously shown that Mps1 is the major kinase activity that copurifies with Dsn1-Flag in their purification scheme. They now investigate the effect of adding ATP and thereby allowing Mps1 phosphorylation in the reconstituted system. They show that addition of ATP decreases the rupture force of kinetochore-microtubule attachments, meaning it weakens the strength of the attachment. This effect can be negated either by inhibiting Mps1 with reversine, or by providing kinetochores in which the Mps1 phosphorylation sites on Ndc80 (most of them in the N-terminal tail) have been mutated to alanine. Thus, like the activity of Ipl1, Mps1 phosphorylation of the Ndc80 N-tail (which is known to be important for full MT affinity) weakens kinetochore-microtubule attachments.

Cellular experiments demonstrate that non-Mps1 phosphorylatable Ndc80 14-A mutants have a functional mitotic checkpoint (contrary to previous claims by Kemmler et al., 2009), but show synthetic sickness with *stu2* alleles that are involved in error correction.

****Major points:****

Within the framework of this experimental setting, the study as presented is logical and clear. The conclusions regarding the effect of Mps1 in this reconstituted system are overall well supported by the data. I have a couple of major and some minor points that can further improve data interpretation and should therefore be considered:

1. In previous publications (e.g. Gutierrez et al., Current Biology 2020), the authors have reported that the Dam1 complex, an established Mps1 substrate, is required for full attachment strength in this system. Are the effects of Mps1-dependent Ndc80 phosphorylation and Dam1 independent from one another? For example would *dad1-1* or non Cdk1 phosphorylatable Dam1 complex further reduce the rupture force in ATP? Or does Mps1 phosphorylation affect, for example, Dam1 binding to Ndc80?

Response: To better understand the effects of ATP treatment, we analyzed the levels of Dam1 on the kinetochores after ATP treatment and did not see any change. We will add this data to a supplemental figure. Dam1 clearly makes a major contribution to the strength of the kinetochores because their strength even after ATP-treatment is higher than the rupture force of kinetochores purified from a *dad1-1* mutant strain. However, as we report in the paper, blocking the eight Mps1 target sites in the tail of Ndc80 was sufficient to block the effect of ATP, so it is unlikely that phosphorylation of the Dam1 complex by Mps1 makes a major contribution to the ATP-dependent kinetochore weakening *in vitro*. We think Dam1 phosphorylation by Aurora B probably contributes independently to error correction, because the *dam1-3D* mutant, carrying phospho-mimetic substitutions in three Aurora B sites, is synthetically lethal when combined with the *ndc80-8D* phospho-mimetic mutant in eight Mps1 sites.

We will add this genetic interaction data to the revised manuscript to provide additional information about the pathways.

2. What is the effect of ATP on initial binding events? Are there differences in the fraction of beads that spontaneously attach laterally at the start of the experiment?

This may allow to draw conclusions whether any kind of binding or specifically force-generating end-on attachments are affected by ATP.

Response: We did measure a reduction in the fraction of free kinetochore-decorated beads capable of binding microtubules upon exposure to ATP (from 20% binding in the absence of adenosine to 11% in the presence of ATP). This observation suggests that the microtubule-binding activity of the kinetochores, like their rupture strength, is reduced upon exposure to ATP, as reported in the methods, in the "rupture force measurements" section. However, because we worked with a low density of kinetochores on the beads, the initial numbers of beads that spontaneously attached was quite low and free beads capable of binding to microtubules were relatively rare. In addition, when we find a bead already attached to the lattice, we cannot distinguish whether it bound initially to the lattice or instead bound to a tip that then grew beyond the bead. For these reasons, we feel it would be very difficult using our current approach to draw statistically significant conclusions about whether there were ATP-dependent changes in the relative affinities of the kinetochores for lateral versus tip attachments.

3. Ndc80-8D has low attachment strength, consistent with lowered MT affinity of the phospho-mimetic Ndc80 tail. Interestingly, Supplementary Figure S4B shows that the amount of Cse4 in the pull-down western appears substantially reduced in 8D vs 8A or wt. Is the amount of co-purified inner kinetochore affected in this mutant? This may be an alternative explanation for decreased attachment strength, for example if the fraction of "full" or "complete" kinetochores may be reduced. Could this also happen upon inclusion of ATP?

Response: The reviewer is correct that the level of Cse4 and other inner kinetochore components is slightly reduced in the Ndc80-8D kinetochores, for reasons that are not clear to us. However, the incubation of wild type kinetochores with ATP does not affect the levels of these proteins, suggesting that the weakened rupture strength is not due to reduced levels of these inner kinetochore proteins. We will add the data showing that ATP does not affect levels of inner kinetochore proteins into a supplemental figure to clarify this point.

****Minor points:****

page 13 (heading): "Weakening occurs via phosphorylation...". Probably good to mention what is weakened ("Weakening of kinetochore-microtubule attachments occurs via phosphorylation...").

Response: We will alter the heading as suggested.

page 14/Figure5C: Median Rupture Force for Ndc80-8D is 4.8 pN according to the text. In the graph it looks like >5 pN.

Response: We thank the reviewer for noticing this mistake and will correct the median rupture force to 5.6 pN.

page 23: comma missing between T21 S37 and T47 (should be T21, S37 and T47)

Response: We thank the reviewer for noticing this omission and will correct it.

page 24/25: different spelling of G1 (sometimes with subscript)

Response: We thank the reviewer for noticing this inconsistency and will correct all to be G1.

page 24/25: ug instead of μg

Response: Thanks. We will fix this mistake.

page 28: Figure 5B instead of Figure 5A

Response: Thanks for noticing this mistake. We will correct this.

Figure 6A: Lambda-Phosphatase treatment for 20 minutes according to figure legend and 30 minutes according to Material and Methods section.

Response: The material and methods section specified a 20-minute incubation with phosphatase, in agreement with the figure legend. We believe the reviewer might have accidentally confused the time value with the temperature, which was 30 degrees.

Figure 6E: One should not draw any conclusions from the anti-phospho T47 blot here, the quality is simply too poor to allow a statement regarding an *mps1-1* effect

Response: While the immunoblots with the T74 phospho-specific antibody are not as clean as many standard antibodies, we have reproduced the results multiple times and therefore feel comfortable concluding that there is a decrease in signal that is Mps1-dependent.

Figure 6: Labelling T47P misleading (Proline substitution?, use pT47 instead)

Response: We will change the labeling on this figure, as suggested, from T74P to pT74. To be consistent, we will also change this nomenclature in the text.

Figure 6F: Make clear in the labelling that a *stu2-AID* background is used here, makes it easier to understand why Auxin is used here.

Response: We will change the labeling, as suggested, to include the genotype of *stu2-AID* in the figure.

how specific is reversine for yeast Mps1? I have not seen any data on this in previous publications.

Response: Reversine is not necessarily specific for Mps1. However, the only kinase activity that co-purifies with the isolated kinetochores is from Mps1, so reversine should inhibit only Mps1 in our *in vitro* experiments. Nevertheless, to further address this concern, we will include optical trapping results using *mps1-1* mutant kinetochores in the revised manuscript. We have already performed these

additional experiments and found that *mps1-1* kinetochores do not undergo ATP-dependent weakening, strongly reinforcing our conclusion that Mps1 is the major kinase involved.

additional genetic interactions might be informative, if Ndc80-8D has weakened attachments, it may have synthetic effects with other mutants (*dam1?*), conversely, *ndc80-8A* may show genetic interactions with *ipl1* alleles, for example.

Response: We agree that the *ndc80* phospho-mutant alleles might have genetic interactions with other mutants. Consistent with this prediction, we have found that *ndc80-8D* is synthetically lethal when combined with the *dam1-3D* mutant in three *ipl1* sites. As mentioned above, we will add this data into the revised text. We will also perform additional genetic interaction experiments with *ipl1* and *mps1* alleles and add any additional interactions we discover into the revised text.

Reviewer #1 (Significance (Required)):

The study adds to the characterization of the effects of Mps1 kinase on kinetochore-microtubule attachments and characterizes the cellular phenotypes of non-Mps1 phosphorylatable Ndc80 mutants. The major conceptual point that Mps1 phosphorylation can weaken kinetochore-microtubule interactions and thereby contributes to error correction in a manner similar to *ipl1* has previously been made in the literature. Maure et al., (Tanaka lab, 2007, Current Biology) have characterized the effects of *mps1* mutant alleles on biorientation of authentic chromosomes and on replicated/unreplicated mini-chromosomes. In particular the experiments with unreplicated mini-chromosomes have revealed less frequent detachment in *mps1* mutants, demonstrating that Mps1 activity is required to release attachments that are not under tension.

Another benefit of this study is that it puts the Kemmler 2009 EMBO J. paper into perspective and corrects some of its claims. In particular the notion of sustained checkpoint activation in the Mps1 phospho-mimetic Ndc80-14D mutant, whose lethality was claimed to be rescued by checkpoint deletion. It is confirmed here that the allele is lethal but cannot be alleviated by simultaneous checkpoint deletion. Conversely, the Ndc80-14A mutant is shown to have a functional checkpoint. One could argue that since the publication of the Kemmler paper, the idea of requirement of Mps1 phosphorylation on Ndc80 for checkpoint activity has not gained any traction in the field, but it's still useful for the field to put some of these earlier claims into perspective. The paper will therefore be interesting to researchers working on mechanisms of chromosome segregation and error correction.

From my background I cannot comment on technical details of the biophysical force spectroscopy experiments (laser trapping), but I have no reason to doubt that the authors accurately report their findings.

Response: We sincerely thank the reviewer for their careful reading, helpful comments, and enthusiasm for our manuscript.

Reviewer #2 (Evidence, reproducibility and clarity (Required)):

This paper focusses on the mechanisms underlying chromosome biorientation in mitosis, an essential process that warrants equal chromosome segregation to the dividing cells. Correction of improper kinetochore-microtubule attachments relies on two conserved protein kinases, Aurora B and Mps1, that detach kinetochores that are not under tension in order to provide them with a second opportunity to

establish bipolar connections. In vivo, Aurora B and Mps1 have intertwined functions and share some common targets. For this reason, despite the large body of literature on the subject, their precise roles in chromosome biorientation have been difficult to tease apart.

The authors take advantage of an in vitro reconstitution assay that they previously published (Akyioshi et al., 2010) to identify the critical target(s) of Mps1 in weakening kinetochore-microtubule connections. The assay uses kinetochore particles purified from budding yeast cells that bear Mps1 but are notably deprived of Aurora B. Upon addition of ATP to activate the co-purified kinases (e.g. Mps1), kinetochores are added to coverslip-anchored microtubules to which they attach laterally. Through a laser trap, kinetochores are brought to the microtubule plus-end and pulled with increasing force until the kinetochore detaches, which allows measurements of the average rupture forces that reflect the strength of the attachments. The approach is straightforward and potentially very powerful, first because it provides a simplified experimental set-up in comparison to the cellular context, and second because it directly measures the impact of protein phosphorylation on the strength of attachments.

The authors convincingly show that Mps1-dependent phosphorylation of the N-terminal part of Ndc80 significantly weakens the strength of kinetochore-microtubule attachments in vitro, while phosphorylation of other known Mps1 targets, such as Spc105, does not seem to have an effect. Eight phosphorylation sites in Ndc80, which were previously identified as Mps1-dependent phosphorylation sites (Kemmler et al., 2009), are shown to be critical to destabilise kinetochore-microtubule attachments in the in vitro reconstitution assays. The authors also present evidence for a moderate involvement of Ndc80 phosphorylation by Mps1 in correcting improper attachments in vivo, suggesting that additional mechanisms are physiologically relevant for error correction.

The experiments are mostly well designed, the data are solid and support the main conclusions. However, to my opinion additional experiments could be performed, as outlined below, to strengthen the physiological relevance of the main findings and corroborate some of the conclusions.

****Major points:****

1. Given the partially overlapping function of Mps1 and Ipl1 (Aurora B) in error correction, the *ndc80-8A* mutant should display synthetic growth and chromosome mis-segregation defects with *ipl1* temperature-sensitive alleles. Conversely, the *ndc80-8D* mutant should suppress the lethality at high temperatures of *mps1-3* mutant cells, which were recently shown to be defective in chromosome biorientation (Benzi et al., 2020). Finally, chromosome mono-orientation could become apparent in *ndc80-8A* cells upon a transient treatment with microtubule-depolymerising drugs, which should amplify the cellular need for error correction.

Response: We agree that further exploration of the possible genetic interactions might help to reinforce the physiological relevance of our main findings. Toward this goal, we will obtain the *mps1-3* mutant to determine whether *ndc80-8D* can suppress its lethality and will add this to the revised manuscript if there is a positive result. As mentioned in response to Reviewer 1, we will add a synthetic lethal interaction between *ndc80-8D* and a *dam1-3D* mutant where the Aurora B sites are altered to the revised text. We will also perform additional genetic interactions with *ipl1* and *mps1* mutants and add any we find into the revision. As requested, we will perform a nocodazole wash out experiment, to determine if *ndc80-8A* cells show a defect in error correction and add this data to the revision if there is a defect.

2. The authors show that Mps1-dependent phosphorylation of Ndc80 is not involved in the spindle assembly checkpoint, a conclusion that contradicts a previous report (Kemmler et al., 2009). They also find, in contrast with the same report, that the lethal phenotype of the *ndc80-14D* phospho-mimetic mutant cannot be rescued by disabling the spindle checkpoint. In my opinion, Kemmler et al. convincingly showed, through a number of different experimental approaches, that *ndc80-14D* cells die because of spindle checkpoint hyperactivation. Not only deletion of checkpoint genes was shown to rescue the lethality, but re-introduction of a wild type copy of the deleted checkpoint gene reinstated lethality. Thus, the explanation invoked here that spontaneous suppressing mutations could underlie the viability of *ndc80-14D* SAC-deficient mutants is not consistent with the published observations. A thorough examination by the authors of the phenotype of *ndc80-14D* cells in their hands should be carried out to support these conflicting conclusions. If authors find that *ndc80-14D* cells actually die because of chromosome mono-orientation, then this would highlight an important function for some or all the six additional phosphorylation sites, relative to the *ndc80-8D* mutant, for chromosome biorientation in vivo.

Response: We were unable to reproduce the data that deletion of the spindle checkpoint suppresses lethality of the *ndc80-14D* mutant, so it remains unclear why our results differ from those of the Kemmler paper. However, we note that re-introducing a wild-type checkpoint gene via transformation and restoring lethality to the *ndc80-14D* cells does not necessarily mean there were no suppressors. While that is one possible interpretation, another possibility is that there was a suppressor mutation in the viable *ndc80-14D* cells that also required the lack of the checkpoint to live. Kemmler and co-workers selected for viability on FOA media and never backcrossed those viable strains to show that they could regenerate the double mutant through a cross with the expected segregation pattern of two mutations, which would have been a more rigorous demonstration that the viability was specifically due to *ndc80-14D* and the checkpoint mutation. Instead, they transformed a wild-type copy of the checkpoint gene back into the strain that was selected for growth on FOA and showed that it reverted the phenotype. This approach cannot rule out a suppressor mutation that fails to suppress in the presence of an active checkpoint. Therefore, in our opinion, the Kemmler paper does not make an entirely convincing case that the *ndc80-14D* cells die because of spindle checkpoint hyperactivation.

To further analyze the phenotype of *ndc80-14D* cells, we have constructed an *Ndc80-AID ndc80-14D* strain and added auxin, to deplete the wild-type copy of Ndc80. In agreement with the findings of Kemmler et al., this did trigger the spindle assembly checkpoint. However, when we made an *Ndc80-AID ndc80-14D mad2* strain and analyzed segregation, we found that chromosome 8 missegregated in 28% of the cells compared to 2% of control cells. This observation suggests that there is a kinetochore defect in these cells that may have triggered the checkpoint and is inconsistent with the mutant solely activating the checkpoint in the absence of any other kinetochore defect. In addition, the levels of Ndc80-14D as well as Mps1 were altered on the mutant kinetochores. The combination of these defects strongly suggests that the *ndc80-14D* mutant alters kinetochore function in addition to leading to constitutive checkpoint signaling. Because our manuscript is mainly focused on phosphorylation of the Mps1 target sites within the N-terminal tail, we do not plan to add this data involving many additional sites, including Ipl1 target sites and sites on the CH domains of Ndc80, into the current manuscript. We will further pursue the other phosphorylation sites in the future.

3. The conclusion that Spc105 phosphorylation by Mps1 is not required for the Mps1-mediated weakening of kinetochore attachments in vitro is based on the comparison between kinetochore particles bearing wild type, untagged Spc105 and particles bearing non-phosphorylatable Spc105-6A

tagged at the C-terminus with twelve myc epitopes. Thus, the presence of the tag could obliterate the effects of the mutations in the phosphorylation sites by destabilising kinetochore-microtubule attachments in the presence of ATP. Consistent with this conclusion, Spc105-6A-12myc-bearing kinetochores withstand lower rupture forces than Spc105-bearing kinetochores upon ATP addition. Furthermore, Spc105-6A-12myc kinetochore particles show an interacting protein at MW above 150 KD that is not present in wild type particles (Fig. S2A), suggesting that either the tag or the mutations might affect kinetochore composition. Thus, this set of experiments should be repeated using Spc105-6A kinetochore particles lacking the tag.

Response: If we understand correctly, the reviewer is suggesting that the myc tag on Spc105-6A could cause an ATP-dependent effect on kinetochore strength. While this is formally possible, it seems highly unlikely to us, for two reasons: First, a myc tag is not expected to bind nucleotides, and while it can sometimes have a general effect on protein stability or interfere with protein-protein interactions, we are not aware of any evidence for a myc tag directly causing an ATP-dependent effect *in vitro*. Second, when we measured Spc105-6A kinetochores in control experiments, without adenosine or with ADP, their rupture strengths were high like wild-type kinetochores. The strength of ADP-treated Spc105-6A kinetochores (8.7 pN), for example, was statistically indistinguishable from that of ADP-treated wild-type kinetochores (8.7 pN, $p = 0.27$ based on a log-rank test). The wild-type-like behavior of untreated and mock-treated Spc105-6A kinetochores indicates that their composition is not affected in a manner that significantly impacts kinetochore-microtubule strength.

4. In general, it would have been informative to complement the data presented here with a mass spec analysis of the composition of kinetochore particles, at least for the experiments that are most relevant to the conclusions. For instance, the composition of the Ndc80-8A kinetochore particles is assumed to be similar to that of wild type kinetochores based on gel silver staining (Fig. S4A; note also that ndc80-8A particles are compared to ndc80-8D particles and not to wild type particles). However, the authors previously showed that kinetochore particles purified from dad1-1 mutant cells (affecting the Dam1 complex) have an apparently identical composition to particles purified from wild type cells by silver staining, yet they display significantly lower resistance to the rupture strength *in vitro* (Akyiوشي et al., 2010). What is the status of the Dam1 complex (or other kinetochore subunits) in kinetochores purified from ndc80-8A/-8D or spc105-6A cells relative to wild type kinetochore particles?

Response: We agree that further characterization of the kinetochore particle composition would be valuable and propose to further analyze the composition by purifying wild-type, Ndc80-8A, Ndc80-8D and Spc105-6A kinetochores and performing immunoblotting against the Dam1 complex. In addition, we will analyze the Ndc80-8A and Ndc80-8D kinetochores by mass spectrometry and report a qualitative analysis of the relative amounts of each kinetochore subcomplex in the revised manuscript supplementary data.

****Minor comment:****

I believe that the right reference for the sentence in the Discussion "If Aurora B is defective, for example, the opposing phosphatase PP1 prematurely localizes to kinetochores" is Liu et al. 2010.

Response: We had cited the reference showing this effect in yeast, since our work was performed in yeast. We will also add the Liu et al paper, which showed the same result in human cells.

Reviewer #2 (Significance (Required)):

Although the experiments are well designed and the conclusions are mainly supported by the data, the question arises as to what extent the *in vitro* assays recapitulate, at least partly, what happens *in vivo*. An emblematic example is the involvement of Spc105 in the error correction pathway. The Biggins lab previously showed that Spc105 phosphorylation by Mps1 and subsequent Bub1 recruitment is not only essential for the spindle assembly checkpoint, but is also crucial for chromosome segregation *in vivo*, as shown by slow-growth phenotype and aneuploidy of the *spc105-6A* non-phosphorylatable mutant (London et al., 2012). Additionally, a recent paper showed that Spc105 is a crucial Mps1 target in chromosome biorientation (Benzi et al., 2020).

In sharp contrast, the *ndc80-8A* mutant, which *in vitro* completely erases the ability of Mps1 to destabilise kinetochore-microtubule attachments, displays no growth defects in otherwise wild type cells and only modestly enhances chromosome mis-segregation in a mutant affecting an intrinsic correction pathway (*stu2ccΔ*). The N-terminal part of Ndc80 (aa 1-116) containing the aforementioned eight phosphorylation sites can even be deleted altogether without any consequence on cell viability (Kemmler et al., 2009). Thus, although the *in vitro* assays presented here produced clear-cut and reproducible results, their physiological relevance *in vivo* remains unclear.

Left apart this criticism, the manuscript has several merits outlined above and will be of interest for people working in the fields of chromosome segregation, kinetochore assembly, spindle assembly checkpoint, etc.

Expertise of this reviewer: mitosis and related checkpoints

Response: We are grateful to the reviewer for carefully reading our manuscript and detailing their concerns. We agree that it can be challenging to establish the physiological relevance of experiments performed *in vitro*. However, our *in vitro* approach allowed the effects of Mps1 specifically on kinetochore-microtubule attachment strength to be disentangled from its numerous other effects *in vivo*. In our view, the relatively mild phenotypes associated with mutants in the Mps1 phosphorylation sites on the Ndc80 tail are consistent with similarly mild phenotypes of mutants in the Aurora B phosphorylation sites on the Ndc80 tail. In both cases, this appears to be due to additional error correction pathways that compensate *in vivo*.

Reviewer #3 (Evidence, reproducibility and clarity (Required)):

Sarangapani, Koch, Nelson et al. applied a combination of *in vitro* biophysical assays with purified kinetochore particles and *in vivo* analyses to investigate the contribution of Mps1 kinase to kinetochore-microtubule (KT-MT) attachment stability and error correction.

The manuscript is well written and the authors nicely highlight the facts that 1) the focus of the field has long been on the contribution of Aurora kinases (Ipl1 in budding yeast) to attachment stability and error correction, and 2) it has been difficult to assess the relative contributions of Aurora versus Mps1 kinases in cell-based experiments. The authors note that their KT particle assay is uniquely positioned to address this gap in our understanding and to specifically isolate the contribution of Mps1 to attachment stability *in vitro*. The findings are well-presented and quite convincing although I have several comments that should be addressed to strengthen the central conclusion that this work has isolated the contribution of Mps1 in their assays.

****Major points:****

1) I think it is important to note that reversine is not specific for Mps1 kinase - although it is typically presented as such in the field. It was initially identified as an Aurora kinase inhibitor (IC50: ~25nM (Aurora B) - 900nM (Aurora A)) that turned out to be an even more potent Mps1 inhibitor (IC50 ~6nM). I have concerns that the *in vitro* assays were done with 5 μ M reversine - a concentration so high that it could certainly inhibit any Ipl1 that is present (see comment 3 below) and possibly even inhibit Bub1 activity as Santaguida et al. (JCB, 2010) measured an IC50 >1 μ M for Bub1 inhibition. It is important to complement/confirm the chemical inhibitor experiment by repeating the rupture assays +/- ATP in KT particles purified from the *mps1-1* strain (shown in Figure 6).

Response: We agree that reversine is not necessarily specific for Mps1 and this concern was also brought up by Reviewer 1. Because Mps1 is the only kinase activity that co-purifies with the isolated kinetochore particles, we expect reversine to inhibit only Mps1 in our *in vitro* assays. However, to further address this point, we will add rupture force assays using kinetochores purified from *mps1-1* mutant cells to the revised manuscript. We have already performed these experiments and they confirm that kinetochores lacking Mps1 do not undergo ATP-dependent weakening. We did not put this data into the original submission because the experiment needs to be performed differently due to altered Dam1 levels. But we will clarify the changes in the materials and methods and add the data to a supplementary figure.

2) If the ATP-mediated reduction in rupture force is lost in the *mps1-1* KT particles, which will also lack Bub1 kinase, then preserving the ATP-dependent reduction in rupture force from KT particles purified from the Bub1 Δ mutant strain would be strong evidence that the contribution of Mps1 kinase has been disentangled from other kinases in this assay.

Response: Although Mps1 recruits Bub1, we think it is unlikely that we are assaying Bub1 kinase activity in our *in vitro* experiments. We cannot detect Bub1 activity on the purified kinetochores using a sensitive radioactive kinase assay (London et al, *Curr Bio* 2011), and the levels of Bub1 in our kinetochore purifications are very low (for example, see Akiyoshi et al, *Nature*, 2010). However, we agree with the reviewer that this caveat should be mentioned and will add this point to the revised text for clarity.

3) Recent work has shown that Sli15-Ipl1 interacts with and is recruited to KTs by the COMA complex (Rodriguez et al., *Curr Biol*, 2019 and Fischbock-Halwachs et al., *eLife* 2019) and that this population of Ipl1 is important for accurate chromosome segregation as also shown 10 years prior by Knockleby and Vogel (*Cell Cycle*, 2009). I realize that this group previously showed (London et al., *Curr Biol*, 2012) that phosphorylation of KT particles was not affected when purified from the *ipl1-321* mutants, but in light of the recent findings how sure are the authors that there is not any Sli15-Ipl1 in the preparations? I think commenting on this would be worthwhile.

Response: We have not detected Ipl1 or Sli15 in the numerous mass spectrometry experiments we have performed on the kinetochore purifications. In addition, we have been separately assaying the effects of Ipl1 phosphorylation on kinetochores for another project (de Regt, <https://doi.org/10.1101/415992>), which independently confirmed that the only detectable kinase activity in our kinetochore purifications is Mps1. We will add this additional reference to the manuscript.

4) Since the interplay between Mps1 and Aurora B are central to this story, the authors should expand upon the sentence on page 5 reading "While there is some evidence that Mps1 regulates Aurora B activity (Jelluma et al., 2010; Saurin et al., 2011; Tighe et al., 2008), significant data suggests it has an independent role in error correction and acts downstream of Aurora B (Hewitt et al., 2010; Maciejowski et al., 2010; Maure et al., 2007; Meyer et al., 2013; Santaguida et al., 2010)." I am not entirely convinced that the in vivo experiments presented here differentiate as to whether Mps1 is upstream from Ipl1 or whether they are acting independently? For example, phosphorylation of T74 looks to be completely lost in figure 6E (although it's difficult to tell since the blot for T74P is very smeary). If they are acting independently in error correction then Ipl1 should still be able to phosphorylate T74 in this condition. However, if the P-T74 really is lost completely in the mcd1-1 cells then this suggests to me that Ipl1 is downstream of Mps1 in this live cell error correction assay.

Response: We thank the reviewer for bringing this to our attention. We did not mean to imply that Mps1 is downstream from Aurora B in budding yeast and were intending only to summarize findings from the literature regarding other organisms. We will revise this section of the text to make that point clearer, and we agree that the order of events remains unresolved. In addition, we will note that Mps1 does not eliminate the phosphorylation detected by the T74 antibody in the revision, to avoid misconceptions about the order of events.

****Other points:****

1) On p.8 "a median strength of 7.5 pN, similar to untreated and ADP-treated kinetochores". Similar is vague so I'm curious as to whether there a statistically significant difference between this and the 9.8 pN and 8.7 pN measured in the other conditions. If so this could be explained by partial dephosphorylation with the phosphatase.

Response: The quoted phrase refers to the 7.5-pN strength measured when λ -phosphatase was included together with ATP (data from Fig. 1D and Supp. Fig. S1B). P-values computed from comparisons of survival plots using the log-rank test show that this strength was not significantly different from the ADP-treated wild-type (8.7 pN, $p = 0.06$), nor was it significantly different from the ADP- and $MnCl_2$ -treated wild-type (8.1 pN, $p = 0.35$). However, it was barely significantly different from $MnCl_2$ -treated wild-type (8.6 pN, $p = 0.03$), and it was more significantly different from untreated wild-type (9.8 pN, $p = 0.0007$). With the revised manuscript, we will include a supplemental table with p-values computed from log-rank tests for all the key statistical comparisons, including those mentioned here.

2) On p.19 the authors note that Aurora A phosphorylates Ndc80 tail during mitosis. Ye et al. (Curr Biol, 2015) also showed that Aurora A can phosphorylate Aurora B sites and that this activity "converges" at the tail to weaken attachments during error correction.

Response: We will add the reference and thank the reviewer for pointing out this omission.

3) Optional: I am curious as to whether the addition of ATP to the Ndc80-8D particles further reduces the rupture force. If so then other sites may also be in play.

Response: We agree this is an interesting question but we have not yet performed those assays and agree it might be worthwhile for a future study.

4) Please comment on why MnCl₂ is used in the rupture assays in Figure S1. I saw no mention of this in the main text.

Response: We include MnCl₂ in the assay because it is required for phosphatase activity and will add this point to the legend of supplementary Figure S1.

5) Consider moving S2 A and B to Figure 3 C and D. This is an interesting result and would go well in the main figure next to the significantly reduced rupture force measurements for the 6A mutant so the reader doesn't have to dig into the supplemental for the data providing this reasonable explanation for the rupture force result.

Response: We thank the reviewer for this suggestion and will move S2A and S2B into Figure 3.

Reviewer #3 (Significance (Required)):

The significance of this relates to focusing on an important phenomenon - error correction - and in looking beyond the traditional focus of the field on Aurora kinases to Mps1 kinase, which is largely implicated in checkpoint signaling. Disentangling the contributions of these two players is an important advance.

The work will be of interest to audiences interested in: kinases, cell division, checkpoints, kinetochore biology, biophysics

The above areas of interest overlap with my expertise.

Response: We thank the reviewer for their enthusiasm for our experiments that help distinguish kinase activities and thus contribute to understanding the process of error correction.

July 26, 2021

Re: JCB manuscript #202106130T

Dr. Sue Biggins
Fred Hutchinson Cancer Research Center
Division of Basic Sciences P.O. Box 19024
1100 Fairview Avenue North, A2-168
Seattle, WA 98109-1024

Dear Dr. Biggins,

Thank you for submitting your manuscript entitled "Kinetochore-associated Mps1 regulates the strength of kinetochore-microtubule attachments via Ndc80 phosphorylation." Please accept our apologies for the delay in the processing of your manuscript, the journal has been understaffed for a long time leading to delays in processing.

We have now had a chance to assess the comments of the Review Commons referees and agree with their assessment that your study represents an important advance in our understanding of kinetochore to microtubule attachments and chromosome segregation. We therefore invite you to submit a full revision as outlined in your response to the reviewer comments.

GENERAL GUIDELINES:

Text limits: Character count for an Article is < 40,000, not including spaces. Count includes title page, abstract, introduction, results, discussion, acknowledgments, and figure legends. Count does not include materials and methods, references, tables, or supplemental legends.

Figures: Articles may have up to 10 main text figures. Figures must be prepared according to the policies outlined in our Instructions to Authors, under Data Presentation, <https://jcb.rupress.org/site/misc/ifora.xhtml>. All figures in accepted manuscripts will be screened prior to publication.

Supplemental information: There are strict limits on the allowable amount of supplemental data. Articles may have up to 5 supplemental figures. Up to 10 supplemental videos or flash animations are allowed. A summary of all supplemental material should appear at the end of the Materials and methods section.

As you may know, the typical timeframe for revisions is three to four months. However, we at JCB realize that the implementation of measures to limit spread of COVID-19 also pose challenges to scientific researchers. Therefore, JCB has waived the revision time limit. We recommend that you reach out to the editors to discuss an appropriate time frame for resubmission if you will need more time. Please note that your revised manuscript will be sent for re-review to the same reviewers that evaluated it for Review Commons. Additionally, JCB policy is that papers are generally considered through only one major revision cycle, so any revised manuscript will likely be either accepted or rejected.

Thank you for this interesting contribution to Journal of Cell Biology. You can contact us at the journal office with any questions, cellbio@rockefeller.edu or call (212) 327-8588.

Sincerely,

Aaron Straight, PhD
Monitoring Editor
Journal of Cell Biology

Dan Simon, PhD
Scientific Editor
Journal of Cell Biology

We are grateful to the reviewers for their thoughtful comments and have performed the following experiments or made the clarifications listed below (blue) in the revised manuscript.

Reviewer #1 (Evidence, reproducibility and clarity (Required)):

The authors use a combination of Dsn1-Flag kinetochore purification from yeast extracts and laser trapping experiments (as in a number of previous studies), to study the effect of Mps1-dependent phosphorylation on reconstituted kinetochore-microtubule attachments *in vitro*. They complement this analysis with genetic experiments characterizing the effects of non-Mps1 phosphorylatable mutants on checkpoint activity and chromosome segregation in yeast.

The authors had previously shown that Mps1 is the major kinase activity that copurifies with Dsn1-Flag in their purification scheme. They now investigate the effect of adding ATP and thereby allowing Mps1 phosphorylation in the reconstituted system. They show that addition of ATP decreases the rupture force of kinetochore-microtubule attachments, meaning it weakens the strength of the attachment. This effect can be negated either by inhibiting Mps1 with reversine, or by providing kinetochores in which the Mps1 phosphorylation sites on Ndc80 (most of them in the N-terminal tail) have been mutated to alanine. Thus, like the activity of Ipl1, Mps1 phosphorylation of the Ndc80 N-tail (which is known to be important for full MT affinity) weakens kinetochore-microtubule attachments.

Cellular experiments demonstrate that non-Mps1 phosphorylatable Ndc80 14-A mutants have a functional mitotic checkpoint (contrary to previous claims by Kemmler et al., 2009), but show synthetic sickness with *stu2* alleles that are involved in error correction.

****Major points:****

Within the framework of this experimental setting, the study as presented is logical and clear. The conclusions regarding the effect of Mps1 in this reconstituted system are overall well supported by the data. I have a couple of major and some minor points that can further improve data interpretation and should therefore be considered:

1. In previous publications (e.g. Gutierrez et al., Current Biology 2020), the authors have reported that the Dam1 complex, an established Mps1 substrate, is required for full attachment strength in this system. Are the effects of Mps1-dependent Ndc80 phosphorylation and Dam1 independent from one another? For example would *dad1-1* or non Cdk1 phosphorylatable Dam1 complex further reduce the rupture force in ATP? Or does Mps1 phosphorylation affect, for example, Dam1 binding to Ndc80?

Response: To better understand the effects of ATP treatment, we analyzed the levels of the Dam1 complex on the kinetochores after ATP treatment and did not see any change. We have added this data to the revised manuscript (Figure S1D, Ask1 levels). Dam1 clearly makes a major contribution to the strength of the kinetochores because their strength even after ATP-treatment is higher than the rupture force of kinetochores purified from a *dad1-1* mutant strain. However, as we report in the paper, blocking the eight Mps1 target sites in the tail of Ndc80 was sufficient to block the effect of ATP, so it is unlikely that phosphorylation of the Dam1 complex by Mps1 makes a major contribution to the ATP-dependent kinetochore weakening *in vitro*. We think Dam1 phosphorylation by Aurora B probably contributes independently to error correction, because the *dam1-3D* mutant, carrying phospho-mimetic substitutions in three Aurora B sites, is synthetically lethal when combined with the *ndc80-8D* phospho-

mimetic mutant in eight Mps1 sites. We have added this genetic interaction data to the revised manuscript to provide additional information about the pathways (Figure 7).

2. What is the effect of ATP on initial binding events? Are there differences in the fraction of beads that spontaneously attach laterally at the start of the experiment?

This may allow to draw conclusions whether any kind of binding or specifically force-generating end-on attachments are affected by ATP.

Response: We did measure a reduction in the fraction of free kinetochore-decorated beads capable of binding microtubules upon exposure to ATP (from 20% binding in the absence of adenosine to 11% in the presence of ATP). This observation suggests that the microtubule-binding activity of the kinetochores, like their rupture strength, is reduced upon exposure to ATP, as reported in the methods, in the "rupture force measurements" section. However, because we worked with a low density of kinetochores on the beads, the initial numbers of beads that spontaneously attached was quite low and free beads capable of binding to microtubules were relatively rare. In addition, when we find a bead already attached to the lattice, we cannot distinguish whether it bound initially to the lattice or instead bound to a tip that then grew beyond the bead. For these reasons, we feel it would be very difficult using our current approach to draw statistically significant conclusions about whether there were ATP-dependent changes in the relative affinities of the kinetochores for lateral versus tip attachments.

3. Ndc80-8D has low attachment strength, consistent with lowered MT affinity of the phospho-mimetic Ndc80 tail. Interestingly, Supplementary Figure S4B shows that the amount of Cse4 in the pull-down western appears substantially reduced in 8D vs 8A or wt. Is the amount of co-purified inner kinetochore affected in this mutant? This may be an alternative explanation for decreased attachment strength, for example if the fraction of "full" or "complete" kinetochores may be reduced. Could this also happen upon inclusion of ATP?

Response: We have repeated purifications of kinetochores from WT, Ndc80-8A and Ndc80-8A cells to better characterize the composition of the kinetochores by mass spectrometry. We have replaced the information in Figure S4 with the new silver stained gels of the purifications and the corresponding mass spectrometry data. This analysis indicates that the inner kinetochore is not affected by the Ndc80 mutations. In addition, the incubation of wild type kinetochores with ATP does not affect the levels of inner kinetochore proteins (Figure S1D), strongly suggesting that the weakened rupture strength is not due to reduced levels of these inner kinetochore proteins.

****Minor points:****

page 13 (heading): "Weakening occurs via phosphorylation...". Probably good to mention what is weakened ("Weakening of kinetochore-microtubule attachments occurs via phosphorylation...").

Response: We altered the heading as suggested (p. 13).

page 14/Figure5C: Median Rupture Force for Ndc80-8D is 4.8 pN according to the text. In the graph it looks like >5 pN.

Response: We thank the reviewer for noticing this mistake and corrected the median rupture force to 5.6 pN in the text (p. 14).

page 23: comma missing between T21 S37 and T47 (should be T21, S37 and T47)

Response: We thank the reviewer for noticing this omission and corrected it (p. 23).

page 24/25: different spelling of G1 (sometimes with subscript)

Response: We thank the reviewer for noticing this inconsistency and corrected all to be G1 (p. 24 and 25).

page 24/25: ug instead of μg

Response: Thanks. We fixed this mistake (p. 24-25).

page 28: Figure 5B instead of Figure 5A

Response: Thanks for noticing this mistake. We corrected it.

Figure 6A: Lambda-Phosphatase treatment for 20 minutes according to figure legend and 30 minutes according to Material and Methods section.

Response: The material and methods section specified a 20-minute incubation with phosphatase, in agreement with the figure legend. We believe the reviewer might have accidentally confused the time value with the temperature, which was 30 degrees.

Figure 6E: One should not draw any conclusions from the anti-phospho T47 blot here, the quality is simply too poor to allow a statement regarding an mps1-1 effect

Response: While the immunoblots with the T74 phospho-specific antibody are not as clean as many standard antibodies, we have reproduced the results multiple times and therefore feel comfortable concluding that there is a decrease in signal that is Mps1-dependent.

Figure 6: Labelling T47P misleading (Proline substitution?, use pT47 instead)

Response: We have changed the labeling on this figure, as suggested, from T74P to pT74. To be consistent, we also changed this nomenclature throughout the text.

Figure 6F: Make clear in the labelling that a *stu2-AID* background is used here, makes it easier to understand why Auxin is used here.

Response: This data has been moved to Figure 7B where we also changed the labeling, as suggested, to include the genotype of *stu2-AID* in the figure.

how specific is reversine for yeast Mps1? I have not seen any data on this in previous publications.

Response: Reversine is not necessarily specific for Mps1. However, the only kinase activity that co-purifies with the isolated kinetochores is from Mps1, so reversine should inhibit only Mps1 in our *in vitro*

experiments. Nevertheless, to further address this concern, we have included optical trapping results using *mps1-1* mutant kinetochores (now reported in Figure S2A). We found that *mps1-1* kinetochores do not undergo ATP-dependent weakening, strongly reinforcing our conclusion that Mps1 is the major kinase involved.

additional genetic interactions might be informative, if Ndc80-8D has weakened attachments, it may have synthetic effects with other mutants (*dam1?*), conversely, *ndc80-8A* may show genetic interactions with *ipl1* alleles, for example.

Response: We agree that the *ndc80* phospho-mutant alleles might have genetic interactions with other mutants. Consistent with this prediction, we have found that *ndc80-8D* is synthetically lethal when combined with the *dam1-3D* mutant in three *Ipl1* sites. As mentioned above, we have added this data into the revised text (Figure 7). We performed additional genetic interaction experiments with *ipl1* and *ndc80-8A* but did not detect an interaction, consistent with the tail domain of Ndc80 being non-essential in yeast.

Reviewer #1 (Significance (Required)):

The study adds to the characterization of the effects of Mps1 kinase on kinetochore-microtubule attachments and characterizes the cellular phenotypes of non-Mps1 phosphorylatable Ndc80 mutants. The major conceptual point that Mps1 phosphorylation can weaken kinetochore-microtubule interactions and thereby contributes to error correction in a manner similar to *Ipl1* has previously been made in the literature. Maure et al., (Tanaka lab, 2007, Current Biology) have characterized the effects of *mps1* mutant alleles on biorientation of authentic chromosomes and on replicated/unreplicated mini-chromosomes. In particular the experiments with unreplicated mini-chromosomes have revealed less frequent detachment in *mps1* mutants, demonstrating that Mps1 activity is required to release attachments that are not under tension.

Another benefit of this study is that it puts the Kemmler 2009 EMBO J. paper into perspective and corrects some of its claims. In particular the notion of sustained checkpoint activation in the Mps1 phospho-mimetic Ndc80-14D mutant, whose lethality was claimed to be rescued by checkpoint deletion. It is confirmed here that the allele is lethal but cannot be alleviated by simultaneous checkpoint deletion. Conversely, the Ndc80-14A mutant is shown to have a functional checkpoint. One could argue that since the publication of the Kemmler paper, the idea of requirement of Mps1 phosphorylation on Ndc80 for checkpoint activity has not gained any traction in the field, but it's still useful for the field to put some of these earlier claims into perspective. The paper will therefore be interesting to researchers working on mechanisms of chromosome segregation and error correction.

From my background I cannot comment on technical details of the biophysical force spectroscopy experiments (laser trapping), but I have no reason to doubt that the authors accurately report their findings.

Response: We sincerely thank the reviewer for their careful reading, helpful comments, and enthusiasm for our manuscript.

Reviewer #2 (Evidence, reproducibility and clarity (Required)):

This paper focusses on the mechanisms underlying chromosome biorientation in mitosis, an essential

process that warrants equal chromosome segregation to the dividing cells. Correction of improper kinetochore-microtubule attachments relies on two conserved protein kinases, Aurora B and Mps1, that detach kinetochores that are not under tension in order to provide them with a second opportunity to establish bipolar connections. In vivo, Aurora B and Mps1 have intertwined functions and share some common targets. For this reason, despite the large body of literature on the subject, their precise roles in chromosome biorientation have been difficult to tease apart.

The authors take advantage of an in vitro reconstitution assay that they previously published (Akyiوشي et al., 2010) to identify the critical target(s) of Mps1 in weakening kinetochore-microtubule connections. The assay uses kinetochore particles purified from budding yeast cells that bear Mps1 but are notably deprived of Aurora B. Upon addition of ATP to activate the co-purified kinases (e.g. Mps1), kinetochores are added to coverslip-anchored microtubules to which they attach laterally. Through a laser trap, kinetochores are brought to the microtubule plus-end and pulled with increasing force until the kinetochore detaches, which allows measurements of the average rupture forces that reflect the strength of the attachments. The approach is straightforward and potentially very powerful, first because it provides a simplified experimental set-up in comparison to the cellular context, and second because it directly measures the impact of protein phosphorylation on the strength of attachments.

The authors convincingly show that Mps1-dependent phosphorylation of the N-terminal part of Ndc80 significantly weakens the strength of kinetochore-microtubule attachments in vitro, while phosphorylation of other known Mps1 targets, such as Spc105, does not seem to have an effect. Eight phosphorylation sites in Ndc80, which were previously identified as Mps1-dependent phosphorylation sites (Kemmler et al., 2009), are shown to be critical to destabilise kinetochore-microtubule attachments in the in vitro reconstitution assays. The authors also present evidence for a moderate involvement of Ndc80 phosphorylation by Mps1 in correcting improper attachments in vivo, suggesting that additional mechanisms are physiologically relevant for error correction.

The experiments are mostly well designed, the data are solid and support the main conclusions. However, to my opinion additional experiments could be performed, as outlined below, to strengthen the physiological relevance of the main findings and corroborate some of the conclusions.

****Major points:****

1. Given the partially overlapping function of Mps1 and Ipl1 (Aurora B) in error correction, the *ndc80-8A* mutant should display synthetic growth and chromosome mis-segregation defects with *ipl1* temperature-sensitive alleles. Conversely, the *ndc80-8D* mutant should suppress the lethality at high temperatures of *mps1-3* mutant cells, which were recently shown to be defective in chromosome biorientation (Benzi et al., 2020). Finally, chromosome mono-orientation could become apparent in *ndc80-8A* cells upon a transient treatment with microtubule-depolymerising drugs, which should amplify the cellular need for error correction.

Response: We agree that further exploration of the possible genetic interactions might help to reinforce the physiological relevance of our main findings. Toward this goal, we obtained the *mps1-3* mutant to determine whether *ndc80-8D* can suppress its lethality and found it did not suppress its temperature sensitivity. We did not add this to the text since it was a negative result. However, as mentioned in response to Reviewer 1, we did discover a synthetic lethal interaction between *ndc80-8D* and a *dam1-3D* mutant where the Aurora B sites are altered and have added this data to the revised text (Figure 7B). As mentioned above, we did not detect any additional genetic interactions with *ipl1* and *ndc80-8A*

mutants but a deletion of the Ndc80 N-terminus does not affect cell viability so this result was not surprising. As requested, we also performed a nocodazole wash out experiment comparing the ability of *ndc80-8A* cells to make proper bioriented attachments after spindle disruption and found they are delayed relative to WT cells. We have added this data to Figure 7A in the revised manuscript.

2. The authors show that Mps1-dependent phosphorylation of Ndc80 is not involved in the spindle assembly checkpoint, a conclusion that contradicts a previous report (Kemmler et al., 2009). They also find, in contrast with the same report, that the lethal phenotype of the *ndc80-14D* phospho-mimetic mutant cannot be rescued by disabling the spindle checkpoint. In my opinion, Kemmler et al. convincingly showed, through a number of different experimental approaches, that *ndc80-14D* cells die because of spindle checkpoint hyperactivation. Not only deletion of checkpoint genes was shown to rescue the lethality, but re-introduction of a wild type copy of the deleted checkpoint gene reinstated lethality. Thus, the explanation invoked here that spontaneous suppressing mutations could underlie the viability of *ndc80-14D* SAC-deficient mutants is not consistent with the published observations. A thorough examination by the authors of the phenotype of *ndc80-14D* cells in their hands should be carried out to support these conflicting conclusions. If authors find that *ndc80-14D* cells actually die because of chromosome mono-orientation, then this would highlight an important function for some or all the six additional phosphorylation sites, relative to the *ndc80-8D* mutant, for chromosome biorientation in vivo.

Response: We were unable to reproduce the data that deletion of the spindle checkpoint suppresses lethality of the *ndc80-14D* mutant, so it remains unclear why our results differ from those of the Kemmler paper. However, we note that re-introducing a wild-type checkpoint gene via transformation and restoring lethality to the *ndc80-14D* cells does not necessarily mean there were no suppressors. While that is one possible interpretation, another possibility is that there was a suppressor mutation in the viable *ndc80-14D* cells that also required the lack of the checkpoint to live. Kemmler and co-workers selected for viability on FOA media and never backcrossed those viable strains to show that they could regenerate the double mutant through a cross with the expected segregation pattern of two mutations, which would have been a more rigorous demonstration that the viability was specifically due to *ndc80-14D* and the checkpoint mutation. Instead, they transformed a wild-type copy of the checkpoint gene back into the strain that was selected for growth on FOA and showed that it reverted the phenotype. This approach cannot rule out a suppressor mutation that fails to suppress in the presence of an active checkpoint. Therefore, in our opinion, the Kemmler paper does not make an entirely convincing case that the *ndc80-14D* cells die because of spindle checkpoint hyperactivation.

To further analyze the phenotype of *ndc80-14D* cells, we have constructed an *Ndc80-AID ndc80-14D* strain and added auxin, to deplete the wild-type copy of Ndc80. In agreement with the findings of Kemmler et al., this did trigger the spindle assembly checkpoint. However, when we made an *Ndc80-AID ndc80-14D mad2* strain and analyzed segregation, we found that chromosome 8 missegregated in 28% of the cells compared to 2% of control cells. This observation suggests that there is a kinetochore defect in these cells that may have triggered the checkpoint and is inconsistent with the mutant solely activating the checkpoint in the absence of any other kinetochore defect. In addition, the levels of Ndc80-14D as well as Mps1 were altered on the mutant kinetochores. The combination of these defects strongly suggests that the *ndc80-14D* mutant alters kinetochore function in addition to leading to constitutive checkpoint signaling. Because our manuscript is mainly focused on phosphorylation of the Mps1 target sites within the N-terminal tail, we did not add this data involving many additional sites, including Ipl1 target sites and sites on the CH domains of Ndc80, into the current manuscript. We will further pursue the other phosphorylation sites in the future.

3. The conclusion that Spc105 phosphorylation by Mps1 is not required for the Mps1-mediated weakening of kinetochore attachments *in vitro* is based on the comparison between kinetochore particles bearing wild type, untagged Spc105 and particles bearing non-phosphorylatable Spc105-6A tagged at the C-terminus with twelve myc epitopes. Thus, the presence of the tag could obliterate the effects of the mutations in the phosphorylation sites by destabilising kinetochore-microtubule attachments in the presence of ATP. Consistent with this conclusion, Spc105-6A-12myc-bearing kinetochores withstand lower rupture forces than Spc105-bearing kinetochores upon ATP addition. Furthermore, Spc105-6A-12myc kinetochore particles show an interacting protein at MW above 150 KD that is not present in wild type particles (Fig. S2A), suggesting that either the tag or the mutations might affect kinetochore composition. Thus, this set of experiments should be repeated using Spc105-6A kinetochore particles lacking the tag.

Response: If we understand correctly, the reviewer is suggesting that the myc tag on Spc105-6A could cause an ATP-dependent effect on kinetochore strength. While this is formally possible, it seems highly unlikely to us, for two reasons: First, a myc tag is not expected to bind nucleotides, and while it can sometimes have a general effect on protein stability or interfere with protein-protein interactions, we are not aware of any evidence for a myc tag directly causing an ATP-dependent effect *in vitro*. Second, when we measured Spc105-6A kinetochores in control experiments, without adenosine or with ADP, their rupture strengths were high like wild-type kinetochores. The strength of ADP-treated Spc105-6A kinetochores (8.7 pN), for example, was statistically indistinguishable from that of ADP-treated wild-type kinetochores (8.7 pN, $p = 0.27$ based on a log-rank test). The wild-type-like behavior of untreated and mock-treated Spc105-6A kinetochores indicates that their composition is not affected in a manner that significantly impacts kinetochore-microtubule strength.

4. In general, it would have been informative to complement the data presented here with a mass spec analysis of the composition of kinetochore particles, at least for the experiments that are most relevant to the conclusions. For instance, the composition of the Ndc80-8A kinetochore particles is assumed to be similar to that of wild type kinetochores based on gel silver staining (Fig. S4A; note also that ndc80-8A particles are compared to ndc80-8D particles and not to wild type particles). However, the authors previously showed that kinetochore particles purified from *dad1-1* mutant cells (affecting the Dam1 complex) have an apparently identical composition to particles purified from wild type cells by silver staining, yet they display significantly lower resistance to the rupture strength *in vitro* (Akyiوشي et al., 2010). What is the status of the Dam1 complex (or other kinetochore subunits) in kinetochores purified from *ndc80-8A/-8D* or *spc105-6A* cells relative to wild type kinetochore particles?

Response: We have better characterized kinetochore particle composition by purifying kinetochores from wild-type, *ndc80-8A*, and *ndc80-8D* cells and performing mass spectrometry to analyze the levels of all kinetochore subcomplexes. This information is now reported in Figure S4B and shows that the composition is similar between all three kinetochore preps, including the levels of the Dam1 complex. As requested, we have also further analyzed the status of the Dam1 complex and other kinetochore subcomplexes in the Spc105-6A kinetochores by immunoblotting (Figure 3C) and found normal levels of all proteins.

****Minor comment:****

I believe that the right reference for the sentence in the Discussion "If Aurora B is defective, for example, the opposing phosphatase PP1 prematurely localizes to kinetochores" is Liu et al. 2010.

Response: We had cited the reference showing this effect in yeast, since our work was performed in yeast. We have now added the Liu et al paper, which showed the same result in human cells.

Reviewer #2 (Significance (Required)):

Although the experiments are well designed and the conclusions are mainly supported by the data, the question arises as to what extent the *in vitro* assays recapitulate, at least partly, what happens *in vivo*. An emblematic example is the involvement of Spc105 in the error correction pathway. The Biggins lab previously showed that Spc105 phosphorylation by Mps1 and subsequent Bub1 recruitment is not only essential for the spindle assembly checkpoint, but is also crucial for chromosome segregation *in vivo*, as shown by slow-growth phenotype and aneuploidy of the *spc105-6A* non-phosphorylatable mutant (London et al., 2012). Additionally, a recent paper showed that Spc105 is a crucial Mps1 target in chromosome biorientation (Benzi et al., 2020).

In sharp contrast, the *ndc80-8A* mutant, which *in vitro* completely erases the ability of Mps1 to destabilise kinetochore-microtubule attachments, displays no growth defects in otherwise wild type cells and only modestly enhances chromosome mis-segregation in a mutant affecting an intrinsic correction pathway (*stu2ccΔ*). The N-terminal part of Ndc80 (aa 1-116) containing the aforementioned eight phosphorylation sites can even be deleted altogether without any consequence on cell viability (Kemmler et al., 2009). Thus, although the *in vitro* assays presented here produced clear-cut and reproducible results, their physiological relevance *in vivo* remains unclear.

Left apart this criticism, the manuscript has several merits outlined above and will be of interest for people working in the fields of chromosome segregation, kinetochore assembly, spindle assembly checkpoint, etc.

Expertise of this reviewer: mitosis and related checkpoints

Response: We are grateful to the reviewer for carefully reading our manuscript and detailing their concerns. We agree that it can be challenging to establish the physiological relevance of experiments performed *in vitro*. However, our *in vitro* approach allowed the effects of Mps1 specifically on kinetochore-microtubule attachment strength to be disentangled from its numerous other effects *in vivo*. In our view, the relatively mild phenotypes associated with mutants in the Mps1 phosphorylation sites on the Ndc80 tail are consistent with similarly mild phenotypes of mutants in the Aurora B phosphorylation sites on the Ndc80 tail. In both cases, this appears to be due to additional error correction pathways that compensate *in vivo*.

Reviewer #3 (Evidence, reproducibility and clarity (Required)):

Sarangapani, Koch, Nelson et al. applied a combination of *in vitro* biophysical assays with purified kinetochore particles and *in vivo* analyses to investigate the contribution of Mps1 kinase to kinetochore-microtubule (KT-MT) attachment stability and error correction.

The manuscript is well written and the authors nicely highlight the facts that 1) the focus of the field has long been on the contribution of Aurora kinases (Ipl1 in budding yeast) to attachment stability and error correction, and 2) it has been difficult to assess the relative contributions of Aurora versus Mps1 kinases in cell-based experiments. The authors note that their KT particle assay is uniquely positioned to address

this gap in our understanding and to specifically isolate the contribution of Mps1 to attachment stability *in vitro*. The findings are well-presented and quite convincing although I have several comments that should be addressed to strengthen the central conclusion that this work has isolated the contribution of Mps1 in their assays.

****Major points:****

1) I think it is important to note that reversine is not specific for Mps1 kinase - although it is typically presented as such in the field. It was initially identified as an Aurora kinase inhibitor (IC50: ~25nM (Aurora B) - 900nM (Aurora A)) that turned out to be an even more potent Mps1 inhibitor (IC50 ~6nM). I have concerns that the *in vitro* assays were done with 5 μ M reversine - a concentration so high that it could certainly inhibit any Ipl1 that is present (see comment 3 below) and possibly even inhibit Bub1 activity as Santaguida et al. (JCB, 2010) measured an IC50 >1 μ M for Bub1 inhibition. It is important to complement/confirm the chemical inhibitor experiment by repeating the rupture assays +/- ATP in KT particles purified from the mps1-1 strain (shown in Figure 6).

Response: We agree that reversine is not necessarily specific for Mps1 and this concern was also brought up by Reviewer 1. Because Mps1 is the only kinase activity that co-purifies with the isolated kinetochore particles, we expect reversine to inhibit only Mps1 in our *in vitro* assays. However, to further address this point, we performed rupture force assays using kinetochores purified from *mps1-1* mutant cells and added this data to the revised manuscript (Figure S2A). These experiments confirmed that kinetochores lacking Mps1 do not undergo ATP-dependent weakening.

2) If the ATP-mediated reduction on rupture force is lost in the mps1-1 KT particles, which will also lack Bub1 kinase, then preserving the ATP-dependent reduction in rupture force from KT particles purified from the Bub1delta mutant strain would be strong evidence that the contribution of Mps1 kinase has been disentangled from other kinases in this assay.

Response: Although Mps1 recruits Bub1, we think it is unlikely that we are assaying Bub1 kinase activity in our *in vitro* experiments. We cannot detect Bub1 activity on the purified kinetochores using a sensitive radioactive kinase assay (London et al, *Curr Bio* 2011), and the levels of Bub1 in our kinetochore purifications are very low (for example, see Akiyoshi et al, *Nature*, 2010). However, we agree with the reviewer that this caveat should be mentioned and have therefore added this point to the revised text for clarity (p. 9).

3) Recent work has shown that Sli15-Ipl1 interacts with and is recruited to KTs by the COMA complex (Rodriguez et al., *Curr Biol*, 2019 and Fischbock-Halwachs et al., *eLife* 2019) and that this population of Ipl1 is important for accurate chromosome segregation as also shown 10 years prior by Knockleby and Vogel (*Cell Cycle*, 2009). I realize that this group previously showed (London et al., *Curr Biol*, 2012) that phosphorylation of KT particles was not affected when purified from the *ipl1-321* mutants, but in light of the recent findings how sure are the authors that there is not any Sli15-Ipl1 in the preparations? I think commenting on this would be worthwhile.

Response: We have not detected Ipl1 or Sli15 in the numerous mass spectrometry experiments we have performed on the kinetochore purifications. In addition, we have been separately assaying the effects of Ipl1 phosphorylation on kinetochores for another project (de Regt, <https://doi.org/10.1101/415992>), which independently confirmed that the only detectable kinase activity in our kinetochore purifications is Mps1. We have added this additional reference to the manuscript.

4) Since the interplay between Mps1 and Aurora B are central to this story, the authors should expand upon the sentence on page 5 reading "While there is some evidence that Mps1 regulates Aurora B activity (Jelluma et al., 2010; Saurin et al., 2011; Tighe et al., 2008), significant data suggests it has an independent role in error correction and acts downstream of Aurora B (Hewitt et al., 2010; Maciejowski et al., 2010; Maure et al., 2007; Meyer et al., 2013; Santaguida et al., 2010)." I am not entirely convinced that the in vivo experiments presented here differentiate as to whether Mps1 is upstream from Ipl1 or whether they are acting independently? For example, phosphorylation of T74 looks to be completely lost in figure 6E (although it's difficult to tell since the blot for T74P is very smeary). If they are acting independently in error correction then Ipl1 should still be able to phosphorylate T74 in this condition. However, if the P-T74 really is lost completely in the *mcd1-1* cells then this suggests to me that Ipl1 is downstream of Mps1 in this live cell error correction assay.

Response: We thank the reviewer for bringing this to our attention. We did not mean to imply that Mps1 is downstream from Aurora B in budding yeast and were intending only to summarize findings from the literature regarding other organisms. We have revised this section of the text to make the point that evidence suggests they might act in independent pathways and removed the comment about the order. The lack of Mps1 does not eliminate the phosphorylation detected by the pT74 antibody and the blots in Figure 6B and 6C show this more clearly than the *mcd1-1* time courses. It is tricky to work with the phospho-specific antibody and we have indicated in the revision that there is still phosphorylation present in the *mcd1-1 mps1-1* time course (p. 16).

****Other points:****

1) On p.8 "a median strength of 7.5 pN, similar to untreated and ADP-treated kinetochores". Similar is vague so I'm curious as to whether there a statistically significant difference between this and the 9.8 pN and 8.7 pN measured in the other conditions. If so this could be explained by partial dephosphorylation with the phosphatase.

Response: The quoted phrase refers to the 7.5-pN strength measured when λ -phosphatase was included together with ATP (data from Fig. 1D and Supp. Fig. S1B). P-values computed from comparisons of survival plots using the log-rank test show that this strength was not significantly different from the ADP-treated wild-type (8.7 pN, $p = 0.06$), nor was it significantly different from the ADP- and $MnCl_2$ -treated wild-type (8.1 pN, $p = 0.35$). However, it was barely significantly different from $MnCl_2$ -treated wild-type (8.6 pN, $p = 0.03$), and it was more significantly different from untreated wild-type (9.8 pN, $p = 0.0007$). We have included a supplemental table (supplementary table 1) with p-values computed from log-rank tests for all the key statistical comparisons, including those mentioned here in the revision.

2) On p.19 the authors note that Aurora A phosphorylates Ndc80 tail during mitosis. Ye et al. (Curr Biol, 2015) also showed that Aurora A can phosphorylate Aurora B sites and that this activity "converges" at the tail to weaken attachments during error correction.

Response: We appreciate this reminder and have added the reference (p. 20).

3) Optional: I am curious as to whether the addition of ATP to the Ndc80-8D particles further reduces the rupture force. If so then other sites may also be in play.

Response: We agree this is an interesting question but we have not yet performed those assays and agree it might be worthwhile for a future study.

4) Please comment on why $MnCl_2$ is used in the rupture assays in Figure S1. I saw no mention of this in the main text.

Response: We include $MnCl_2$ in the assay because it is required for phosphatase activity and have added this point to the legend and labeled the graphs in supplementary Figure S1B.

5) Consider moving S2 A and B to Figure 3 C and D. This is an interesting result and would go well in the main figure next to the significantly reduced rupture force measurements for the 6A mutant so the reader doesn't have to dig into the supplemental for the data providing this reasonable explanation for the rupture force result.

Response: We thank the reviewer for this suggestion but have entirely removed this data from the paper. We originally analyzed Mps1 levels on kinetochores purified from strains in which both Spc105 proteins (Spc105 and spc105-6A) were Myc-epitope tagged. However, the optical trapping experiment was performed using kinetochores purified from WT and spc105-6A-myc strains, so we further characterized the composition of these kinetochores. We found that Mps1 levels are similar when these two kinetochore preps are analyzed, so we added these immunoblots to the revised manuscript and removed the other preps (Figure 3C). Given the similar levels of Mps1 on the kinetochores used for the laser trap comparison, it is unclear why ATP addition has a stronger effect on Spc105-6A kinetochores as compared to WT, so we have removed speculation that it might be due to altered Mps1 levels. Despite this uncertainty, the spc105-6A kinetochores treated with ADP behave like WT, so we do not believe that the tag on Spc105 is the reason for the ATP-dependent weakening. In addition, these data clearly indicate that these sites on Spc105 do not contribute to the ATP dependent weakening that we study in this manuscript.

Reviewer #3 (Significance (Required)):

The significance of this relates to focusing on an important phenomenon - error correction - and in looking beyond the traditional focus of the field on Aurora kinases to Mps1 kinase, which is largely implicated in checkpoint signaling. Disentangling the contributions of these two players is an important advance.

The work will be of interest to audiences interested in: kinases, cell division, checkpoints, kinetochore biology, biophysics

The above areas of interest overlap with my expertise.

Response: We thank the reviewer for their enthusiasm for our experiments that help distinguish kinase activities and thus contribute to understanding the process of error correction.

August 31, 2021

RE: JCB Manuscript #202106130R

Dr. Sue Biggins
Fred Hutchinson Cancer Research Center
Division of Basic Sciences P.O. Box 19024
1100 Fairview Avenue North, A2-168
Seattle, WA 98109-1024

Dear Dr. Biggins,

Thank you for submitting your revised manuscript entitled "Kinetochore-bound Mps1 regulates kinetochore-microtubule attachments via Ndc80 phosphorylation." We would be happy to publish your paper in JCB pending final revisions necessary to address minor points from reviewers and to meet our formatting guidelines (see details below).

A. MANUSCRIPT ORGANIZATION AND FORMATTING:

Full guidelines are available on our Instructions for Authors page, <https://jcb.rupress.org/submission-guidelines#revised>. **Submission of a paper that does not conform to JCB guidelines will delay the acceptance of your manuscript.**

- 1) Text limits: Character count for Articles is < 40,000, not including spaces. Count includes title page, abstract, introduction, results, discussion, and acknowledgments. Count does not include materials and methods, figure legends, references, tables, or supplemental legends.
- 2) Figures limits: Articles may have up to 10 main text figures.
- 3) Figure formatting: Scale bars must be present on all microscopy images, including inset magnifications. Molecular weight or nucleic acid size markers must be included on all gel electrophoresis. Please add scale bars to images in Figure 7A,D and MW markers to Figures 3C, 5B, 6A-E, S1D, and S2C.
- 4) Statistical analysis: Error bars on graphic representations of numerical data must be clearly described in the figure legend. The number of independent data points (n) represented in a graph must be indicated in the legend. Statistical methods should be explained in full in the materials and methods. For figures presenting pooled data the statistical measure should be defined in the figure legends. Please also be sure to indicate the statistical tests used in each of your experiments (both in the figure legend itself and in a separate methods section) as well as the parameters of the test (for example, if you ran a t-test, please indicate if it was one- or two-sided, etc.). Also, if you used parametric tests, please indicate if the data distribution was tested for normality (and if so, how). If not, you must state something to the effect that "Data distribution was assumed to be normal but this was not formally tested."

5) Materials and methods: Should be comprehensive and not simply reference a previous publication for details on how an experiment was performed. Please provide full descriptions (at least in brief) in the text for readers who may not have access to referenced manuscripts. The text should not refer to methods "...as previously described."

6) For all cell lines, vectors, constructs/cDNAs, etc. - all genetic material: please include database / vendor ID (e.g., Addgene, ATCC, etc.) or if unavailable, please briefly describe their basic genetic features, even if described in other published work or gifted to you by other investigators. Please be sure to provide the sequences for all of your oligos: primers, si/shRNA, RNAi, gRNAs, etc. in the materials and methods. You must also indicate in the methods the source, species, and catalog numbers/vendor identifiers (where appropriate) for all of your antibodies, including secondary.

7) Microscope image acquisition: The following information must be provided about the acquisition and processing of images:

- a. Make and model of microscope
- b. Type, magnification, and numerical aperture of the objective lenses
- c. Temperature
- d. Imaging medium
- e. Fluorochromes
- f. Camera make and model
- g. Acquisition software
- h. Any software used for image processing subsequent to data acquisition. Please include details and types of operations involved (e.g., type of deconvolution, 3D reconstitutions, surface or volume rendering, gamma adjustments, etc.).

8) References: There is no limit to the number of references cited in a manuscript. References should be cited parenthetically in the text by author and year of publication. Abbreviate the names of journals according to PubMed.

9) Supplemental materials: There are strict limits on the allowable amount of supplemental data. Articles may have up to 5 supplemental figures and 10 videos. Please also note that tables, like figures, should be provided as individual, editable files. A summary of all supplemental material should appear at the end of the Materials and methods section. Please include one brief sentence per item.

10) eTOC summary: A ~40-50 word summary that describes the context and significance of the findings for a general readership should be included on the title page. The statement should be written in the present tense and refer to the work in the third person. It should begin with "First author name(s) et al..." to match our preferred style.

11) Conflict of interest statement: JCB requires inclusion of a statement in the acknowledgements regarding competing financial interests. If no competing financial interests exist, please include the following statement: "The authors declare no competing financial interests." If competing interests are declared, please follow your statement of these competing interests with the following statement: "The authors declare no further competing financial interests."

12) A separate author contribution section is required following the Acknowledgments in all research manuscripts. All authors should be mentioned and designated by their first and middle initials and full surnames. We encourage use of the CRediT nomenclature (<https://casrai.org/credit/>).

13) ORCID IDs: ORCID IDs are unique identifiers allowing researchers to create a record of their various scholarly contributions in a single place. At resubmission of your final files, please consider providing an ORCID ID for as many contributing authors as possible.

B. FINAL FILES:

Thank you for this interesting contribution, we look forward to publishing your paper in Journal of Cell Biology.

Sincerely,

Aaron Straight, PhD
Monitoring Editor
Journal of Cell Biology

Dan Simon, PhD
Scientific Editor

Reviewer #1 (Comments to the Authors (Required)):

The authors largely addressed the concerns I raised when this manuscript was at Review Commons. My major concern related to whether the addition of ATP affected rupture forces for KT particles purified from the *mps1-1* strain. The resubmitted version includes this new data and, importantly, they did not observe ATP-dependent weakening of the rupture forces in the *mps1-1* particles.

I support publication of the work in JCB.

Reviewer #2 (Comments to the Authors (Required)):

In this revised version the authors have addressed most of my concerns and technical suggestions raised after the initial submission. The revised manuscript contains a substantial number of additional experiments that are useful and have improved the study. In particular, with the additional *in-vivo* experiments, the characterization of the non-Mps1 phosphorylatable Ndc80 alleles is connected better to the biophysical *in-vitro* part of the study. In comparison to the previous version, the authors provide information on the relationship between Mps1 regulation and the Dam1-dependent strengthening of attachments *in vitro* (Figure S2). They include characterization of the ATP effect on kinetochore assemblies purified from an *mps1-1* mutant, an important control for the *in-vitro* experiments (Figure S2). There is also more complete information on the composition of the kinetochore assemblies purified from the different mutants (Figure S4), which serves to rule out effects that are not related to outer kinetochore function. Finally, a previous shortcoming of the study was that the Ndc80-8D mutant, which substantially decreases rupture force of the attachment *in vitro*, does not have an obvious phenotype in cells, raising concerns about the physiological relevance of the finding. The authors now show that *ndc80-8D* displays synthetic sickness with *dam1-3D* (Figure 7B), a mutant mimicking Ipl1 phosphorylation of Dam1, an established part of the error correction pathway. Together with the demonstration that *ndc80-8A* cells are delayed in establishing bi-orientation after a noc washout (Figure 7A), this provides evidence that the Mps1 sites in the tail of Ndc80 are indeed relevant for error correction in cells.

Taken together I think the study now clears the bar expected for a JCB paper and I am supportive of publication. As noted in my previous review, the conceptual advance regarding understanding mechanisms or error correction is somewhat modest, but to me this study serves as a careful characterization of Mps1-mediated phosphorylation of these sites on Ndc80, and it balances/corrects previous claims regarding their importance for checkpoint function and kinetochore attachment.

Remaining minor points that should be considered:

Page 20 top: "... produced constitutively weak kinetochores" - one should add "in vitro" to indicate that the "weakness" is derived from the *in-vitro* experiments.

It is mentioned in the introduction, but I think Maure et al., 2007 deserves a bit more consideration in

the discussion, because this paper has demonstrated the role of Mps1 on biorientation in yeast before, with a clever use of mono- and dicentric unreplicated mini-chromosomes.

Reviewer #3 (Comments to the Authors (Required)):

The authors have convincingly addressed my criticisms. The new functional data strengthen the main conclusions of the paper and lend support to the physiological relevance of the main findings for the error correction process. Therefore, I support publication in JCB.